# The influence of extratropical cross-tropopause mixing on the correlation between ozone and sulfate aerosol in the lowermost stratosphere

Philipp Joppe[1,2], Johannes Schneider[2], Katharina Kaiser[1,2], Horst Fischer[3], Peter Hoor[1], Daniel Kunkel[1], Hans-Christoph Lachnitt[1], Andreas Marsing[4], Lenard Röder[3], Hans Schlager[4], Laura Tomsche[1,4], Christiane Voigt[1,4], Andreas Zahn[5], and Stephan Borrmann[1,2]

[1]Institute for Atmospheric Physics, Johannes Gutenberg University Mainz, Germany
[2]Particle Chemistry Department, Max Planck Institute for Chemistry, Mainz, Germany
[3]Atmospheric Chemistry Department, Max Planck Institute for Chemistry, Mainz, Germany
[4]Institute of Atmospheric Physics, Deutsches Zentrum für Luft- und Raumfahrt (DLR), Oberpfaffenhofen, Germany
[5]Karlsruhe Institute of Technology, Institute of Meteorology and Climate Research, Karlsruhe, Germany

**Correspondence:** Philipp Joppe (phjoppe@uni-mainz.de)

**Abstract.** The chemical composition of the upper troposphere/lower stratosphere region (UTLS) is influenced by horizontal transport of air masses, vertical transport within convective systems and warm conveyor belts, rapid turbulent mixing, as well as photochemical production or loss of species. This results in the formation of the extratropical transition layer (ExTL), which is defined by the vertical structure of CO and studied until now mostly by means of trace gas correlations. Here, we extend
the analysis to include aerosol particles and derive the sulfate aerosol to ozone ($O_3$) correlation in Central Europe from aircraft in-situ measurements during the CAFE-EU/BLUESKY mission. The mission probed the UTLS during the COVID-19 period with significant reduced anthropogenic emissions. We operated a compact time-of-flight aerosol mass spectrometer (C-ToF-AMS) to measure the chemical composition of non-refractory aerosol particles in the size range from about 40 to 800 nm. In our study, we find a correlation between the sulfate mass concentration and $O_3$ in the lower stratosphere. The correlation
exhibits some variability exceeding the mean sulfate to ozone correlation over the measurement period. Especially during one flight, we observed enhanced mixing ratios of sulfate aerosol in the lowermost stratosphere, where the analysis of trace gases shows tropospheric influence. However, back trajectories indicate, that no recent mixing with tropospheric air occurred within the last 10 days. Therefore, we analyzed volcanic eruption databases and satellite $SO_2$ retrievals from TROPOMI for possible volcanic plumes and eruptions to explain the high amounts of sulfur compounds in the UTLS. From these analyses and the
combination of precursor and particle measurements, we conclude that gas-to-particle conversion of volcanic $SO_2$ leads to the observed enhanced sulfate aerosol mixing ratios.

## 1 Introduction

The chemical composition of upper tropospheric aerosol particles is highly variable because primary aerosols from natural and anthropogenic ground sources reach this altitude (Martinsson et al., 2019), and secondary aerosols are formed here from

gas-to-particle conversion. However, the stratospheric aerosol composition is less complex as the main component is particulate sulfate ($SO_4^{2-}$) with concentrations between 0.1 and 40 µg m$^{-3}$ accompanied by minor tropospheric compounds (Deshler, 2008; Murphy et al., 2013; Brimblecombe, 2014; Kremser et al., 2016).

The stratospheric aerosol layer, also known as "Junge layer" (Junge and Manson, 1961), is part of the lowermost stratosphere (LMS) and is located roughly between 15 and 25 km (Hofmann and Rosen, 1981). The chemical composition of the aerosol layer underlies seasonal variations induced by the Brewer-Dobson circulation (Martinsson, 2005; Friberg et al., 2014) and volcanic activity, which may increase the aerosol optical depth (AOD) by up to 40 % (Friberg et al., 2018). Sulfate aerosol is formed due to oxidation of carbonyl sulfide (OCS) and sulfur dioxide ($SO_2$) (Crutzen, 1976; Brühl et al., 2012; Solomon et al., 2011; Kremser et al., 2016) and has an average radius in undistorted conditions of 170 nm (e.g., Tilmes and Mills, 2014). Both precursor gases have their main sources in the troposphere. OCS is the main sulfur containing trace gas in the atmosphere with direct emissions by the oceans or biomass burning as well as photochemical production by oceanic emissions like dimethyl sulfide (DMS) or carbon disulfide ($CS_2$) (Andreae, 1990; Brühl et al., 2012; Kremser et al., 2016). $SO_2$ is primary emitted by industrial processes like the fossil fuel burning. While degassing volcanoes contribute to the tropospheric $SO_2$ budget (Voigt et al., 2014), explosive eruptions can directly inject $SO_2$ into the lower stratosphere (Kremser et al., 2016). Another direct sulfate aerosol source are aircraft which emit soot and volatile sulfate containing particles at cruise altitudes into the upper troposphere/lower stratosphere (Voigt et al., 2010; Williamson et al., 2021; Tomsche et al., 2022).

Transport processes of aerosol particles into the UTLS have been subject of several studies, for example with focus on tropical processes or the Asian tropopause aerosol layer (ATAL) (e.g., Appel et al., 2022; Fadnavis et al., 2013; Froyd et al., 2009; Höpfner et al., 2019). Tracer correlations of aerosol particles with ozone ($O_3$) and nitrous oxide ($N_2O$) based on high altitude in-situ measurements also have been used in the context of polar vortex dynamics after a major volcanic eruption (Borrmann et al., 1993, 1995). In those studies, the temporal evolution of the correlation between the mixing ratios of sulfate aerosol, surface area for aerosol particles with a size between roughly 10 nm and more than one µm, in addition to $O_3$ inside and outside of the polar vortex was analyzed over the course of 22 months after the Mt. Pinatubo eruption in 1991. The observations revealed a slow development of a linear correlation between the $O_3$ versus aerosol number and surface mixing ratios in the mid latitude UTLS, which degraded again later on. This demonstrated the suitability of aerosol properties as dynamical tracers once the microphysical processes like new particle formation, coagulation, and condensational growth following a volcanic eruption into the stratosphere have ceased. Furthermore, a temporally stable equilibrium of particle size/surface area has been established.

In the extratropics, the chemical composition of the tropopause region is influenced not only by the Brewer-Dobson circulation, but also by convection, mixing along the subtropical and polar frontal jet streams, breaking of gravity and Rossby waves, and vertical wind shear (e.g., Gettelman et al., 2011; Kaluza et al., 2021). This forms a transition layer above the tropopause, called the extratropical transition layer (ExTL), where tropospheric as well as stratospheric influence is observed (Hoor et al., 2004; Hegglin et al., 2009; Gettelman et al., 2011; Konopka and Pan, 2012; Barré et al., 2013). The effect of these small-scale mixing processes on the chemical composition of aerosol particles in the ExTL is not well known until now (Kunkel et al., 2019). The lifetime of atmospheric aerosol particles with diameters lower than one µm can reach one month or more (Jaenicke,

1980). This is sufficient for the particles to be transported up into the tropopause region over long distances and subsequently, by mixing processes, into the stratosphere. Furthermore, the lifetime of more than a month corresponds to the timescale that gas-to-particle conversion needs to form sulfate aerosol particles from $SO_2$ as a precursor gas in the UTLS (Jurkat et al., 2010; Gorkavyi et al., 2021; Rollins et al., 2017).

In our study, we use tracer-tracer correlations as a tool for mixing diagnostics to identify stratospheric air masses and underlying mixing processes. The basic principle of this method is to use a tropospheric tracer with sources in the troposphere and a rather constant stratospheric background, e.g. carbon monoxide (CO) and water vapor ($H_2O$), and a tracer with only a stratospheric increase or decrease and a fairly constant mixing ratio in the troposphere, e.g. $O_3$ or $N_2O$. In a scatter plot with the tropospheric tracer on the abscissa and the stratospheric tracer on the ordinate, one would expect two separated reservoirs that are not connected if no mixing processes would occur. If mixing takes place, both reservoirs are be connected by mixing lines, where the mixing ratios are between the two regimes, depending on the state of mixing (Fischer et al., 2000; Hoor et al., 2002).

With this study we want to introduce particulate sulfate as stratospheric tracer in the correlation with $O_3$ and find processes that are responsible for the variability of the correlation between sulfate aerosol and $O_3$ mixing ratios. Therefore, we use in-situ aircraft measurements from the CAFE-EU (Chemistry of the Atmosphere Field Experiment over Europe)/BLUESKY mission, conducted in spring 2020 from Oberpfaffenhofen, Germany.

## 2 Methods

### 2.1 Data overview

The CAFE-EU/BLUESKY measurement campaign was conducted with the research aircraft HALO (High Altitude and Long Range Research Aircraft) and DLR-Falcon, both operated by the German Aerospace Center (DLR). The measurement flights were performed over Central Europe and the North Atlantic between 16 May and 09 June 2020, partly co-located with both aircraft (see Fig. 1). The measurements were conducted during the first COVID-19 lockdown in Germany and Europe, such that the main goal of the campaign was to investigate the atmospheric changes during reduced industrial activity and lower emissions compared to other times (Voigt et al., 2022). This point leads to flight planning during the campaign with focus on urban areas and low altitude profiles and less on studying processes in the UTLS region. Therefore, it was not possible to conduct measurements over the complete vertical extent of the ExTL during May 2020. Nevertheless, we were able to obtain measurement data up to 14 km altitude representing the chemical composition of the UTLS. During the campaign period, air traffic was significantly reduced over Europe by up to 80 % (Schumann et al., 2021a, b). Krüger et al. (2022) found substantial reduction in aerosol particles in the lower troposphere in this period and Reifenberg et al. (2022) could explain the observed reduction in some tracer concentrations with the reduced emissions of pollutants. Tomsche et al. (2022) investigated the $SO_2$ concentrations in the UTLS region above Europe, which was influenced by changes in sulfur sources such as aviation as well

as sinks. Here we focus on the transport processes in the extratropical transition layer, which has been probed with a set of instrumentations onboard HALO and DLR-Falcon.

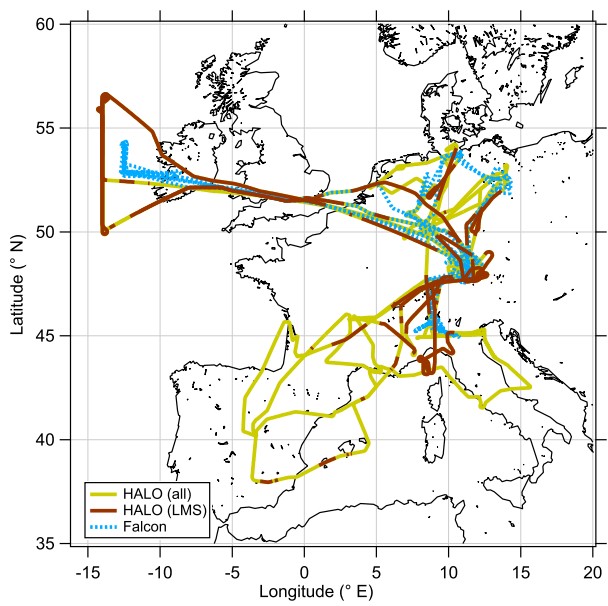

**Figure 1.** Overview map of all measurement flights performed during the CAFE-EU/BLUESKY measurement campaign between 16 May and 09 June 2020. The HALO flight path as solid line divided into the complete data set (yellow) and stratospheric (brown) segments, while the DLR-Falcon flight path is shown as a dashed blue line.

## 2.2 Instrumentation

For our study, we use the chemical composition of aerosol particles measured onboard HALO and trace gas measurements onboard HALO and DLR-Falcon. The chemical composition of non-refractory aerosol particles was measured on HALO with a compact time-of-flight aerosol mass spectrometer (C-ToF-AMS) for particles in the size range between 40 nm and 800 nm (Drewnick et al., 2005; Schulz et al., 2018) and with this within the same size range as previous studies (e.g., Borrmann et al., 1995). We obtain quantitative information on the mass concentration of sulfate, nitrate, ammonium, organic matter and
chloride normalized to STP conditions. The measurement interval of the C-ToF-AMS during the campaign was 30 s, resulting in a spatial resolution of about 6 km in the UTLS region. The accuracy of the AMS is about 30 % (Bahreini et al., 2009; Canagaratna et al., 2007; Middlebrook et al., 2012). In the following, we use the mixing ratio instead of the mass concentration for comparison with the trace gas measurements. Integrated into the C-ToF-AMS, we use an optical particle counter (OPC) manufactured by GRIMM (OPC 1.129) to measure the aerosol size distribution in 31 size channels from 250 nm to larger than
32 μm.

In addition to the aerosol chemical composition and size data, we use trace gas measurements, like $SO_2$, CO, $H_2O$, $O_3$ and

nitric acid ($HNO_3$) onboard HALO and DLR-Falcon.

CO measurements on HALO were performed with the quantum cascade laser absorption spectrometer TRISTAR (Tadic et al., 2017; Röder et al., 2023) with a total measurement uncertainty of 3 % at 10 s time resolution. $O_3$ on HALO was measured by FAIRO which measured on the basis of a UV photometer and chemiluminescence (Zahn et al., 2011). On board the DLR-Falcon, CO and $O_3$ were measured with a cavity ring-down spectrometer (PICARRO G2401) and a dual-cell UV photometer (TE 49C), respectively. Both instruments were calibrated before and after the flights with standards that can be traced back to the GAW Station Hohenpeissenberg. The precision/accuracy of the CO and $O_3$ measurements are 3 ppbv/ 5 ppbv and 3 %/ 5%, respectively. An additional in-situ data set is provided by the atmospheric chemical ionization mass spectrometer (AIMS) deployed on DLR-Falcon and includes information on gaseous $SO_2$ and $HNO_3$. For the detection of upper tropospheric and lower stratospheric $SO_2$ and $HNO_3$ mixing ratios the AIMS uses $SF_5^-$ reagent ions (Voigt et al., 2014; Jurkat et al., 2016; Marsing et al., 2019; Tomsche et al., 2022). The one sigma detection limit is 0.0006 to 0.0017 ppbv and 0.005 to 0.009 ppbv for $SO_2$ and $HNO_3$, respectively. The total uncertainty for $SO_2$ is 22.7 % (Tomsche et al., 2022) and 16 % for $HNO_3$ (Ziereis et al., 2022).

## 2.3 Meteorology and trajectories

In addition to the in-situ measurement data, we use model data interpolated onto the flight path of both aircraft. For meteorological information, we use the ERA5 reanalysis data set with a temporal resolution of six hour and a grid spacing of one degree in the horizontal and a vertical spacing of approximately 500 m in the UTLS (Hersbach et al., 2020). Based on the native variables we additionally calculated potential vorticity (PV) and equivalent latitude. The equivalent latitude is a framework to account for reversible transport under adiabatic conditions and thus get information on potential diabatic transport or mixing. For the calculation, for different isentropes a PV-contour line having the same potential vorticity and potential temperature is transformed into a pole centered circle. The equivalent latitude is the enclosing latitude of this circle (e.g., Lary et al., 1995; Hegglin et al., 2006; Krause et al., 2018). These calculations are done over isentropic surfaces from 240 up to 2000 K from the ERA5 reanalysis data interpolated on potential temperature.

For our analysis of the air mass origin and possible transport pathways, we use trajectories calculated with the Lagrangian analysis tool (LAGRANTO; Sprenger and Wernli, 2015). Therefore, we initialize a set of 231 trajectories every 30 seconds along the flight path. The starting points of each trajectory set are placed in a three-dimensional cross around the initial point of the flight path, to gain a better statistic and to minimize interpolation errors between the measurements and the model grid. More specifically, we take the location of the aircraft and add five additional points every 0.01 degree in all four horizontal directions (north, east, south and west) resulting in 21 points arranged in a cross shape (including the aircraft position/location). This cross pattern of 21 points is repeated in 10 additional vertical levels in one hPa steps, five levels above and five levels below the flight altitude. Thus, we get a total of 231 trajectories starting locations at each release time, providing information for 10 days back in time with quantities such as potential temperature and potential vorticity.

## 3 Results

### 3.1 Part 1: Correlation of particulate sulfate and ozone

$O_3$ is a suitable tracer to identify stratospheric air masses due to the photochemical production of $O_3$ in the stratosphere and its low abundance in the troposphere and its local chemical lifetime of years in the lower stratosphere. There are several ways to identify the tropopause and thus the lower boundary of the stratosphere from measured $O_3$ mixing ratios. For example, fixed threshold values of typically 70 ppbv or 100 ppbv have been used (e.g., Bethan et al., 1996; Staehelin, 2003). This method has the disadvantage to neglect the seasonal cycle of $O_3$ and thus the threshold value can be too low or too high when periods exceeding one month are analyzed. Another way to determine the $O_3$-based tropopause is to take the seasonal cycle into account by using a daily threshold for the $O_3$ tropopause. This method is described in Zahn et al. (2004) and Thouret et al. (2005) on the basis of long-term observations. In our study we use the method of Zahn et al. (2004) to calculate daily $O_3$ thresholds for the tropopause. Figure 2 shows the seasonal cycle of the $O_3$ mixing ratio at the chemical tropopause and the 2 PVU (potential vorticity units, $10^{-6} m^2 s^{-1} K kg^{-1}$) dynamical tropopause, both calculated after Zahn et al. (2004). For the period of our study (16 May to 09 June), we only see small differences in the results between the chemical and the dynamical tropopause since both thresholds are close together around 120 ppbv $O_3$ during the time of our measurements. We extracted all stratospheric data (i.e. all data points with $O_3$ mixing ratios larger than the daily threshold) and calculated the frequency distribution of the potential vorticity from the ERA5 data set (Fig. 2) to verify that the chemical tropopause inferred from the measured $O_3$ mixing ratios and the calculated thresholds correspond to the dynamical tropopause. This shows that the vast majority of the stratospheric data points have PV values larger than 2 PVU. In Fig. 2, we observe two modes in the PV distribution of the stratospheric data which can be explained by the different stratospheric ages of the sampled air masses (Bönisch et al., 2009). The first mode is resulting from air freshly mixed into the stratosphere with PV values close to the dynamical tropopause. The second mode with values larger than 8 PVU describes air originating from the high stratosphere with no tropospheric influence. In total, less than 4 % of the stratospheric data points have PV values below the 2 PVU threshold. Thus we use the definition of the chemical tropopause based on $O_3$ mixing ratios after Zahn et al. (2004) in this study.

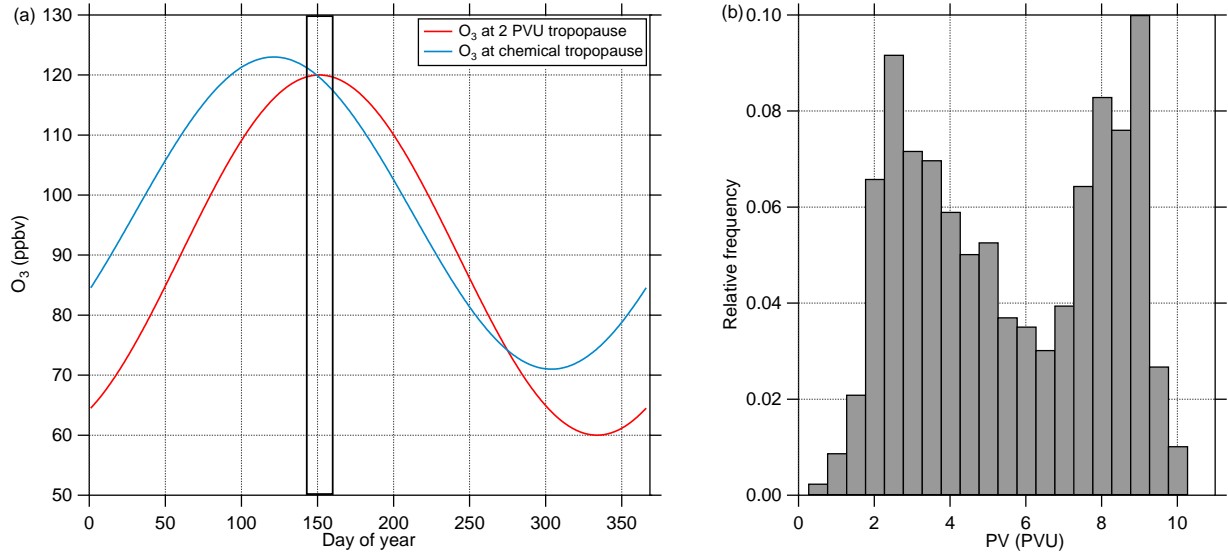

**Figure 2.** (a) Calculated seasonal cycle of $O_3$ mixing ratios at the 2 PVU dynamical tropopause and the chemical tropopause, both calculated after Zahn et al. (2004). The measurement period is marked with the solid frame. (b) Relative frequency of PV values interpolated to the AMS data points from CAFE-EU/BLUESKY identified as stratospheric using the calculated $O_3$ mixing ratios at the chemical tropopause. Less than 4 % of the data points show PV values lower than 2 PVU indicating a negligible amount of tropospheric air. The complete stratospheric data set holds 2049 data points which represent 100 % of this subset.

Besides $O_3$, there are other trace gases used as indicators of stratospheric air masses, e.g. $H_2O$ or $N_2O$. Moreover, within limits, aerosol particle properties can be applied as well (Borrmann et al., 1993, 1995). A good example is the mass con-
160 centration of particulate sulfate, which increases in the stratosphere due to the formation from precursors, which takes 30 to 60 days and reaches its maximum in the Junge layer, where particulate sulfate is present in the form of binary solution droplets with inclusions of sulfuric acid (Junge and Manson, 1961; Brühl et al., 2012; Kremser et al., 2016). Thus, we expect a positive correlation between particulate sulfate and $O_3$ in the stratosphere. The observations made on HALO during the CAFE-EU/BLUESKY campaign confirm this (Fig. 3). Here, two distinct regimes appear in the correlation plot of particulate
sulfate and $O_3$. In the tropospheric regime, the sulfate mass concentration shows a high variability at $O_3$ levels below 100 ppbv, depending on the source regions within the the boundary layer, e.g. industrial areas. In the stratosphere, we observe a linear correlation between the two species with a slope of 900 to 2300 ppbv/ppbv, but with variations between the individual measurement flights, i.e. on short time scales of a few days. Note that the accuracy of the C-ToF-AMS of about 30 % (Bahreini et al., 2009) does not affect the observed different slope regimes in the correlation of sulfate aerosol and ozone, because the
quantities determining the accuracy (ionization efficiency, collection efficiency and inlet transmission efficiency) do not change over the short period of a two-week measurement campaign..

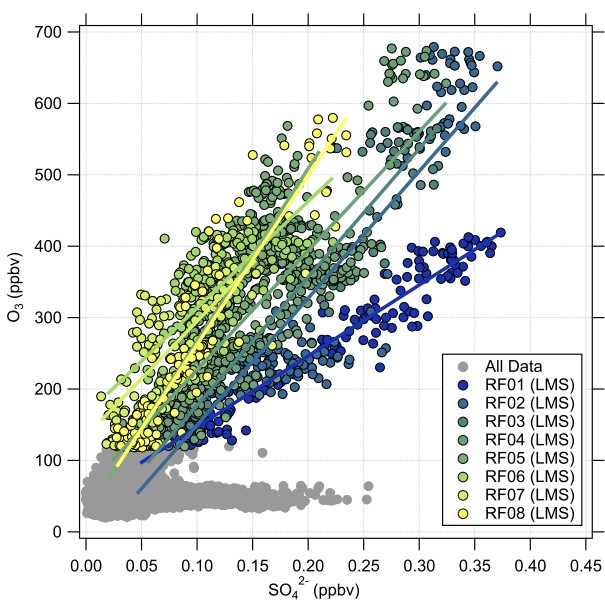

**Figure 3.** Correlation between particulate sulfate mixing ratio and $O_3$ mixing ratio for the full CAFE-EU/BLUESKY data set. The color-coded data points indicate the stratospheric data derived from the chemical tropopause $O_3$ mixing ratios. The grey data shows the complete data set including the tropospheric data. The solid color-coded lines represent the linear regressions for the individual flights.

This analysis shows that the correlation between particulate sulfate and $O_3$ can be used as a tool for analysing air masses in terms of their stratospheric character. The variations in the slope and compactness of the correlation appear on short timescales of a few days. The aim of this study is to understand these variations and link them to possible atmospheric processes. There are a number of different pathways by which sulfur species can be transported from the troposphere into the stratosphere and thus be a possible reason for the observed variability (e.g., Kremser et al., 2016). Feinberg et al. (2019) show the modelled atmospheric sulfur budget under volcanically quiescent conditions and the pathways that lead to the formation of particulate sulfate in the stratosphere. Among these pathways, the most efficient one is the mixing of precursor gases such as OCS and $SO_2$ into the stratosphere, where OCS is oxidised to $SO_2$ and $SO_2$ is further converted to sulfuric acid, forming sulfate aerosol. It is important to emphasise that this budget is valid for volcanically quiescent conditions, because in the presence of volcanic eruptions an additional large source of $SO_2$ adds up to the other pathways. In this case, $SO_2$ is transported in the eruption column up into the free troposphere or even the UTLS region; depending on the strength of the eruption. Then $SO_2$ is converted to sulfuric acid and particulate sulfate also in the upper troposphere or even in the stratosphere (Kremser et al., 2016).

The low sulfate mixing ratios at the chemical tropopause (Fig. 3) show that direct mixing of high sulfate aerosol concentrations from the troposphere to the stratosphere was not observed during the campaign, so some other processes need to be taken into account. This observation of low particulate sulfate aerosol amounts at the chemical tropopause is very robust over the whole campaign period and there it might be controlled by atmospheric processes that need more investigation.

Previous studies with focus on the Raikoke eruption in 2019 determined no significant contribution from this volcanic eruption

(Tomsche et al., 2022; Reifenberg et al., 2022). The following section will show the possible influence of cross-tropopause
mixing, especially of the precursor gas $SO_2$ and the potential influence of a more current volcanic eruption.

### 3.2  Part 2: Case study on aerosol chemical composition related to mixing processes

Volcanic influence is one of the possibilities that can explain the observed variability in the correlation. One major eruption
occurred in 2019 by the Raikoke volcano. However, Tomsche et al. (2022) and Reifenberg et al. (2022) showed that this
eruption does not have a significant impact on the measurements during CAFE-EU/BLUESKY. In the following, we focus on
a case study to explain the variability of the particulate sulfate correlation with $O_3$, especially for research flight RF01, as the
slope in the $O_3$-$SO_4^{2-}$ correlation for this flight clearly differs from the other flights (see Fig. 3).

The anti-correlation between CO and $O_3$ can be used to identify mixing processes between the troposphere and stratosphere
(Fischer et al., 2000; Hoor et al., 2002). The presence of mixing lines connecting the sampled tropospheric and the stratospheric
air masses indicates recent mixing processes. In Fig. 4, most of the stratospheric data of the whole campaign data set lie in a
region of anticorrelated CO and $O_3$. Thus, it can be concluded that the measured air masses in the stratosphere are influenced
by tropospheric air that was mixed across the tropopause. As expected, the correlation in Fig. 4b shows much higher particulate
sulfate mixing ratios at higher $O_3$ levels in the stratosphere and lower mixing ratios close to the tropopause. However, we can
identify an anomaly of high particulate sulfate mixing ratios (yellow points) at about 400 ppbv $O_3$ and 40 ppbv CO (see also
Fig. C1 and Fig. C4). Here, the measured mixing ratios of sulfate aerosol are in the range we would expect higher up in the
stratosphere.

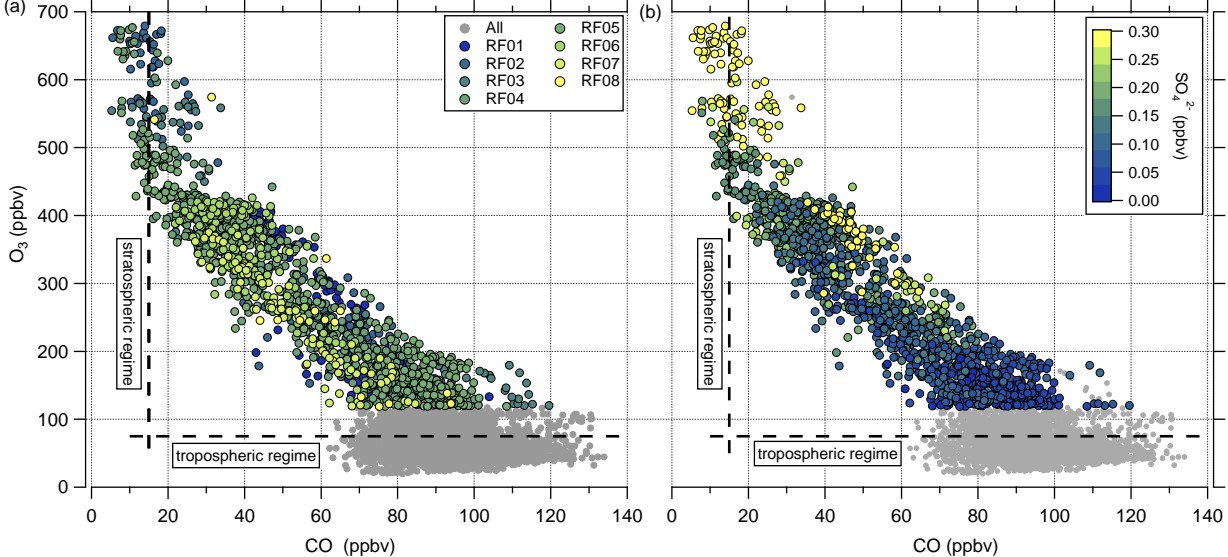

**Figure 4.** (a) CO-$O_3$ correlation for all data points in grey and the stratospheric data color-coded with the flight numbers. (b) The same
correlation but the stratospheric data is color-coded with the sulfate aerosol mixing ratio. In addition, the dashed lines indicate the mean
tropospheric and stratospheric regimes without mixing processes.

In the following, we will analyze whether this anomaly is caused by a cross-tropopause mixing event and if such events can explain the observed variability in the $SO_4$-$O_3$ correlation. For this analysis, we binned our data set along equivalent latitude, which can be used as a dynamical tracer (Butchart and Remsberg, 1986; Hegglin, 2005), and potential temperature to see where the anomaly is located (see Fig. 5). The observed sulfate anomaly occurs in Fig. 5b between 40° N and 45° N at potential temperatures between 345 K and 350 K. It is not connected to the observed stratospheric aerosol layer that starts at higher altitudes, above the 370 K isentrope (see Fig. 5). The potential vorticity indicates that this region is in the vicinity of the jet stream and with this mixing processes might have occur or even be present. The $O_3$ distribution does not show such an anomalous observation like found in the particulate sulfate(Fig. 5a), but we can observe the expected increase from the troposphere to the stratosphere. This location of the anomaly is in good agreement with the previous observation that the anomaly is located on a mixing line in the CO-$O_3$ correlation in a transition regime between the troposphere and the stratosphere.

We further investigate the meteorology over the campaign period to determine whether the anomaly might be influenced by mixing processes. In particular, we use the vertical wind shear $S^2$ and the static stability $N^2$ (see also Appendix B) to identify regions with higher potential for mixing processes. Kaluza et al. (2021) and Kunkel et al. (2019) showed in their study that in regions with high vertical wind shear ($S^2 > 4 \cdot 10^{-4} \text{ s}^{-2}$), conditions are favourable for rapid mixing. Figure 6 shows the analysis of the stability parameters mentioned above along with the resulting gradient Richardson number. The vertical wind shear shows high values in the region of the sulfate anomaly (see Fig. 6b), exceeding the mentioned threshold for enhanced mixing. This also results in a reduction of the gradient Richardson number in the same bin to values close to the critical threshold of 0.25 (see Fig. 6c), which indicates favourable conditions for turbulence. The static stability shows the expected transition from tropospheric to stratospheric values (see Fig. 6a). However, regarding the stability analysis we probed several regions which show favorable conditions for instability and cross-tropopause mixing. Nevertheless, here we focus on the region with the strongest signal, where the observed sulfate anomaly was measured.

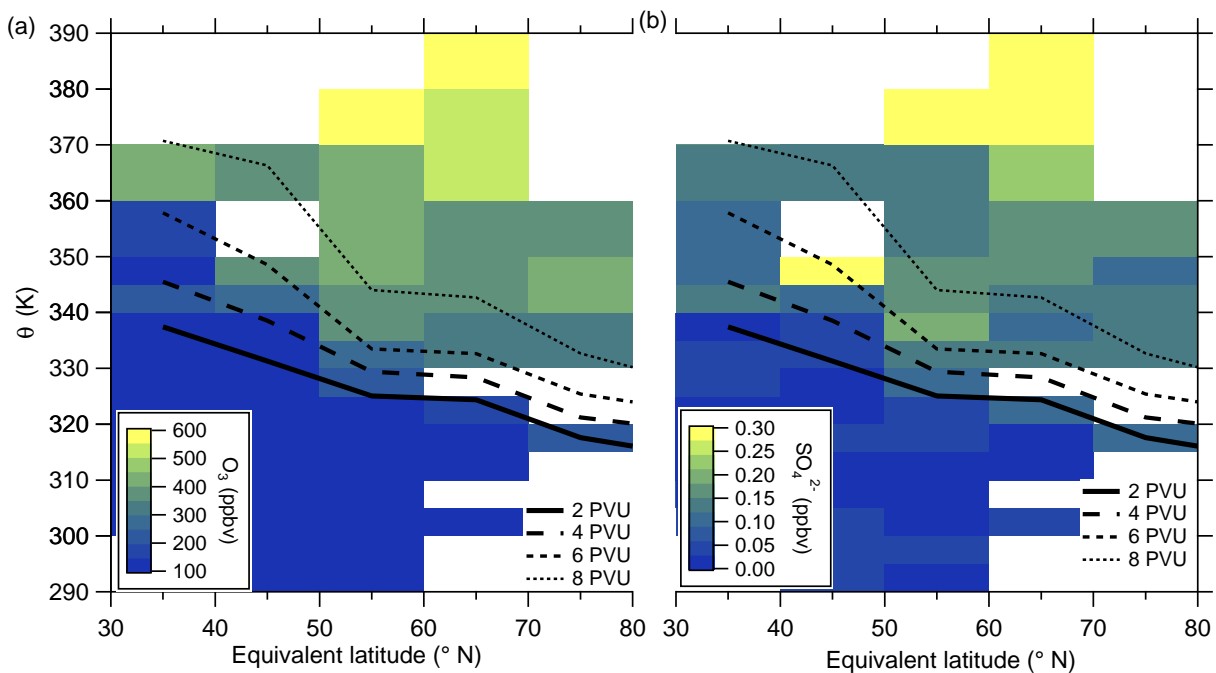

**Figure 5.** Median of $O_3$ mixing ratio (a) and sulfate aerosol mixing ratio (b) in the $\theta$-equivalent latitude space. Below 350 K, we use 5 K vertical resolution and $10°$ equivalent latitude bins. Above 350 K, we enlarge the vertical bins to 10 K to obtain a higher statistical evaluation basis (see Fig. A1). In general, only bins with more than 10 data points are evaluated. We added the 2 to 8 PVU lines to show the location of the dynamical tropopause and the extent of the ExTL into the stratosphere. We only show the region of interest for our study between 30 and 80 degrees north.

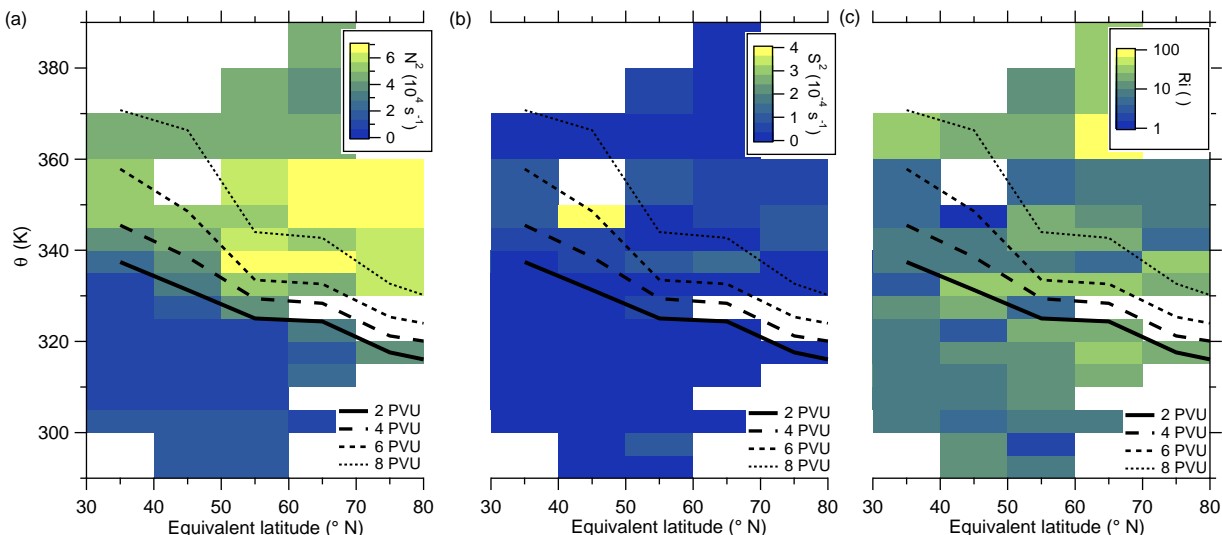

**Figure 6.** Median of static stability $N^2$ (a), vertical wind shear $S^2$ (b) and gradient Richardson number (c) in the $\theta$-equivalent latitude space. Below 350 K, we use 5 K vertical resolution and $10°$ equivalent latitude bins. Above 350 K, we enlarge the vertical bins to 10 K to obtain a higher statistical evaluation basis (see Fig. A1). In general, only bins with more than 10 data points are evaluated. We added the 2 to 8 PVU lines to show the location of the dynamical tropopause and the extent of the ExTL into the stratosphere. We only show the region of interest for our study between 30 and 80 degrees north.

These findings suggest that mixing occurred in this measurement region and is one influence for the variation of the $SO_4$-$O_3$ correlation. To prove this, we investigated the data in the region of the sulfate anomaly in more detail. Therefore, we extracted the data points that contribute to the anomaly. In total there are 45 measurement points by the C-ToF-AMS in the bin between 40 and $50°$ N and $\Theta = 345$ to 350 K. The majority of these points (41 points) was sampled during research flight RF01 in a time span of 20 minutes (see Fig. 3). The flight was conducted on 23 May 2020 over Germany, while the anomaly was measured in an area over Lower Saxony (see also Fig. C1, Fig. C2 and Fig. C4). A time series of the measurements from this flight is shown in Fig. 7 and Fig. E3, while the period of the anomaly is marked with a colored box.

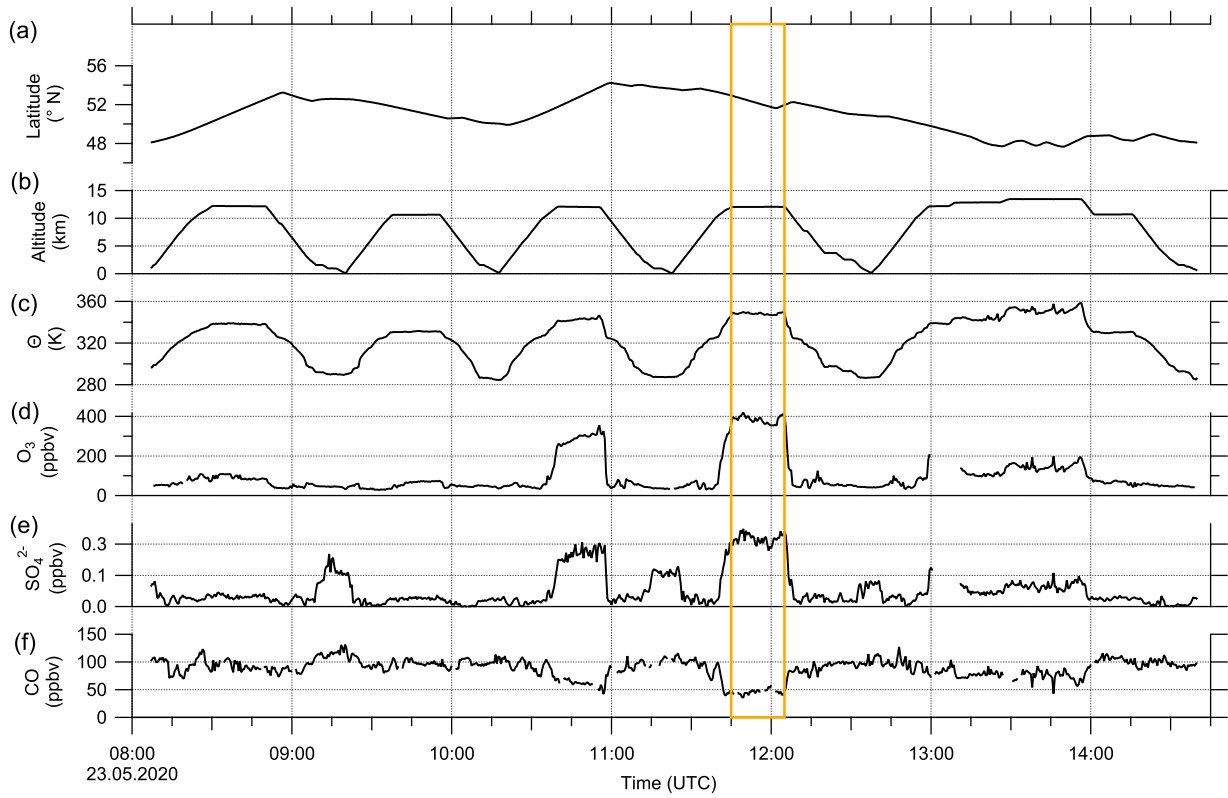

**Figure 7.** Time series of in-situ measurements during research flight RF01 on 23 May 2020 in 30 s time steps. (a) Latitude, (b) altitude, (c) potential temperatur $\theta$, (d) $O_3$ mixing ratios, (e) sulfate aerosol mixing ratios and (f) CO mixing ratios. The orange box marks the period with the observed sulfate anomaly described in the text and shown in Fig. C1. In addition, we added a timeseries of all aerosol species to the appendix (see Fig. E3).

Figure 4 reveals the correlation between CO and particulate sulfate in the lower troposphere, as well as the positive correlation between particulate sulfate and $O_3$ and the anti-correlation between sulfate and CO in the stratosphere. To identify mixing from the measurements, we use two types of scatterplots. The first is the previously introduced scatterplot using CO and $O_3$ with different color coding (Fig. 8a-c), the second is the scatterplot with $H_2O$ and $O_3$ (Hegglin et al., 2009) (see Fig.

8d-f). This figure contains only data measured on 23 May 2020. The $H_2O$-$O_3$ method follows the same principle, with high water vapor mixing ratios in the troposphere and a constant stratospheric background value around five ppmv (Hegglin et al., 2009). In our data set, the lowest observed $H_2O$ values are around 10 ppmv, indicating that we did not fully reach stratospheric background conditions. All of these scatterplots show two separate branches of mixing lines. This feature is most obvious in the $H_2O$-$O_3$-correlation. Here, one mixing line connects the tropopause with around $H_2O$= 40 ppmv and $O_3$= 100 ppbv and

the LMS with decreasing $H_2O$ (down to 10 ppmv) at $O_3$=400 ppbv . This mixing line includes also the measured sulfate anomaly and was observed over Northern Germany (see 7 and C4). The second mixing line is not as pronounced and starts at dryer air masses with $H_2O$ = 20 ppmv and only reaches up to $O_3$=200 ppbv . These observations were made later on the flight

over Southern Germany (see also Fig. 7 and C4).

Regardless of the type of scatterplot, we observe an increase in the particulate sulfate mixing ratio and potential temperature
along the mixing line, starting at the tropopause and extending into the stratosphere (Fig. 8). In contrast, we also observe a
decrease in the CO mixing ratio. This result is consistent with the assumption that tropospheric air enters the stratosphere at
lower potential temperatures with lower amounts of sulfate aerosol, accordingly higher mixing ratios of precursor gases, and
higher CO. In the stratosphere, gas-to-particle conversion of OCS and $SO_2$ will lead to an increase of particulate sulfate. In
contrast to sulfate, CO will decrease during the transport into the stratosphere, both by dilution and photochemical destruction,
with an atmospheric lifetime of one to three months (Seinfeld and Pandis, 2016). This is almost the same time as the calculated
e-folding time of gas-to-particle conversion of $SO_2$ to sulfate aerosol in the mid-latitude LMS region (Jurkat et al., 2010).

Further evidence that the anomaly is caused by tropospheric influence are the lower $O_3$ values and the water vapor mixing
ratios. If the air masses were stratospheric origin, we would expect $O_3$ mixing ratios higher than 400 ppbv and a water vapor
mixing ratio around five ppmv. Instead, we observe lower $O_3$ mixing ratios and water vapor mixing ratios around $10 - 20$
260 ppmv.

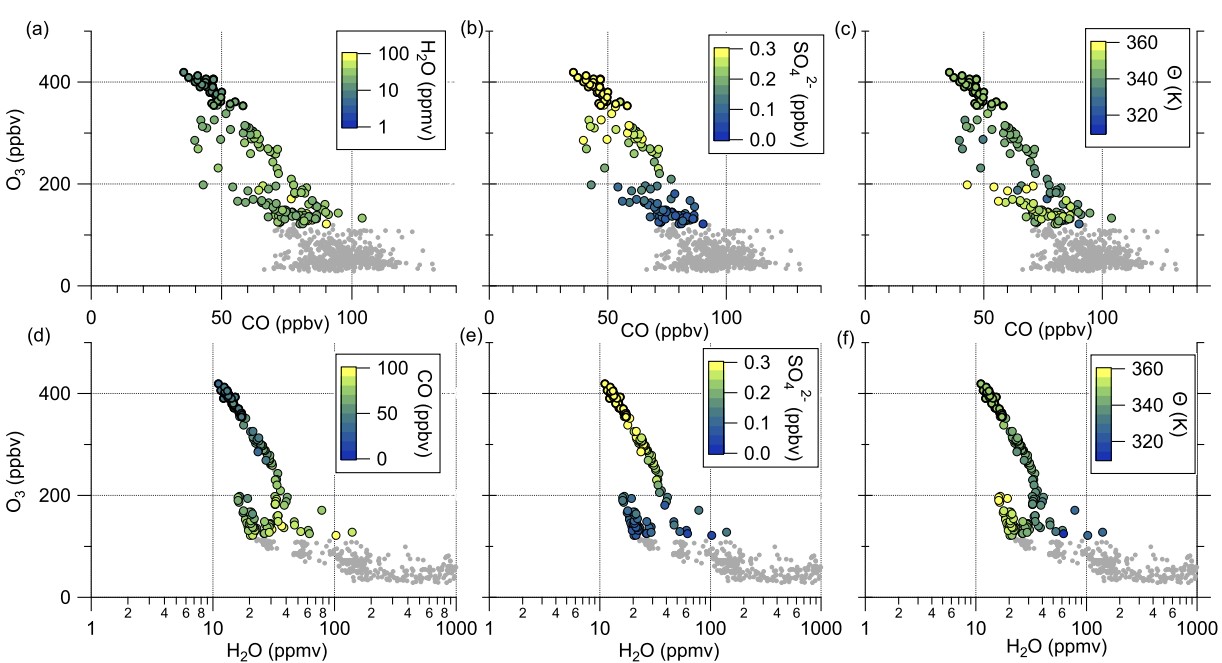

**Figure 8.** Tracer-tracer correlations for 23 May 2020 to identify mixing processes. The CO-$O_3$ correlations are color-coded with (a) $H_2O$,
(b) $SO_4^{2-}$ and (c) $\theta$. The $H_2O$-$O_3$ correlations are color-coded with (d) CO , (e) $SO_4^{2-}$ and (f) $\theta$. The data correspond to the blue colored data
in Fig. 3 and the sulfur anomaly in Fig. 5.

Similarly to the HALO measurements, we analysed measurements conducted on the DLR-Falcon for this analysis. The DLR-
Falcon performed a measurement flight on the same day and sampled in the area where the anomaly was encountered just 40

min later than HALO (Fig. C2), which allows a comparison of the measurements on both platforms with respect to dynamical processes. The DLR-Falcon does not reach the same high altitudes as HALO, so the air masses were probed at lower levels and thus show higher CO mixing ratios. This time, we use the scatterplot of CO and $HNO_3$ to identify mixing in combination with gas-to-particle conversion (Fig. 9b). $HNO_3$ was already introduced as a stratospheric tracer by Proffitt et al. (1989) or Arnold et al. (1989) and utilised because the $O_3$ data for the DLR-Falcon are not available for this flight. Additionally, we added the measurements from HALO to this figure (Fig. 9a) to directly link them to the process of gas-to-particle conversion. We identify a mixing line in the scatterplots, connecting troposphere and stratosphere. The HALO measurements in Fig. 9a show an increase of the total particle number concentration along the mixing line, whereas the $SO_2$ mixing ratios on this mixing line (Fig. 9b) show a reduction with respect to the measured tropospheric maxima of $0.1$ ppbv, which is an indication for gas-to-particle conversion along this mixing line. This conclusion is also supported by the correlation of the total particle number concentration with the individual species of chemical composition, measured by the C-ToF-AMS (see Fig. E5). Here we can see that the particles forming and growing are mainly sulfate aerosol particles, and the particles do not have a pure tropospheric composition. The process of gas-to-particle conversion requires a source of high mixing ratios of precursor gases, in this case $SO_2$. In addition, due to the high solubility of $SO_2$, it requires a very fast and mostly dry process for transport into the UTLS. One possible process that fulfills these conditions is volcanic eruptions, which leads to the assumption that the observed sulfate aerosol particles are most likely formed by volcanic influence.

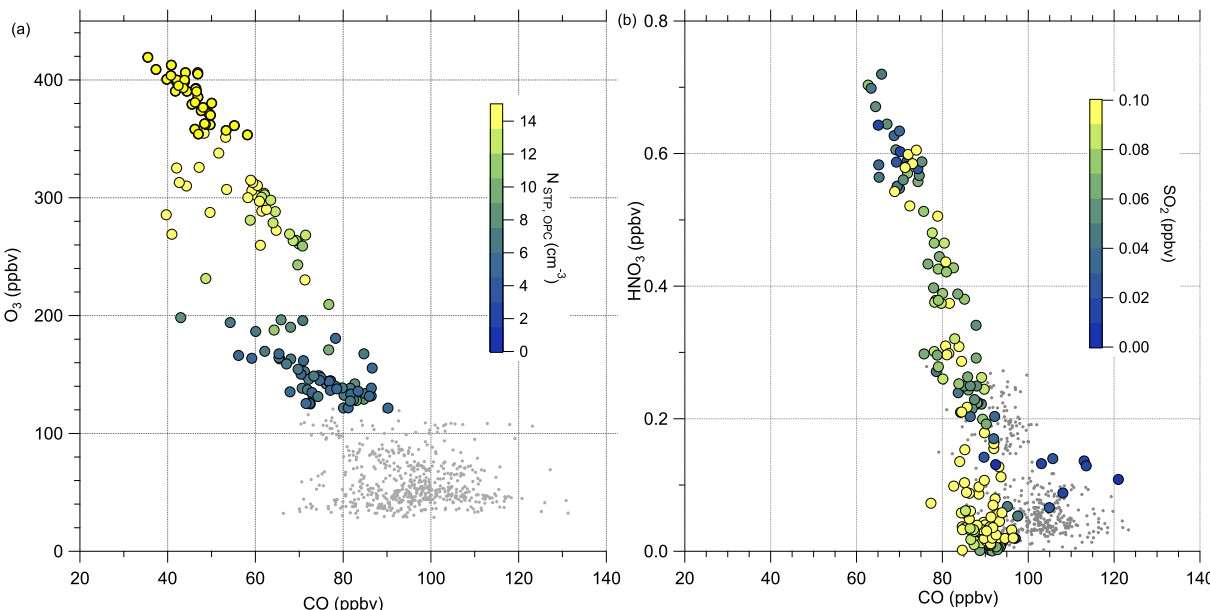

**Figure 9.** Scatterplot of CO and $O_3$ measured on HALO (a) and CO and $HNO_3$ measured on the DLR-Falcon (b) for the investigated flight on 23 May 2020. In (a) the color-code is showing the total particle number concentration by the OPC. For the DLR-Falcon measurements in (b) we color-coded with the $SO_2$ mixing ratio. Both measurement platforms observe a mixing line in the probed area. Whereas the total aerosol number concentration is increasing along this mixing line the $SO_2$ measurements show a reduction. This is one possible indicator for the process of gas-to-particle conversion.

As a complement to the measurement data, we analyzed the meteorological situation along the flight path for the period of
the anomaly (Fig. B1). The flight path was located just above the maximum of the subtropical jet stream and a layer of strong vertical wind shear (Fig. B1a). Further, we see that the flight path crossed a layer of low gradient Richardson number (Fig. B1)b, and later continued slightly above this layer. This indicated a region of instability which is an important factor for mixing processes.
The previous discussion has shown that mixing between tropospheric and stratospheric air masses most likely occurred before
and during the in situ measurements.
Hereafter, we examine the origin of the air masses comprised of high mixing ratios of sulfate aerosol. Therefore, we exploit LAGRANTO backward trajectories starting at a grid around the flight path with 230 trajectories for each measurement point. The trajectories are calculated 10 days backward, to see whether the air masses show any fresh influence from the troposphere or a rather stable flow within the lowermost stratosphere. For our analysis, we filtered the trajectories by selecting only those
with a minimum potential temperature below 345 K and with an increase of at least 5 K potential temperature. Figure 10 shows that the selected trajectories are close together and move with the jet stream. The trajectories cross the region of East Asia within 10 days before the measurement and most of them crossed China and its megacities like Chengdu and Shanghai. The time series of potential temperature shows that no boundary layer air masses were transported to the measurement region in

the last 10 days (Fig. 10a), considering the assumption that the trajectories can resolve convective uplift. As a consequence, the
enhanced particulate sulfate needs to be older than 10 days and was most likely not directly mixed into the LMS as particulate
sulfate and with this also supports the findings in the previous analysis (see e.g. Fig. 9 and E5).

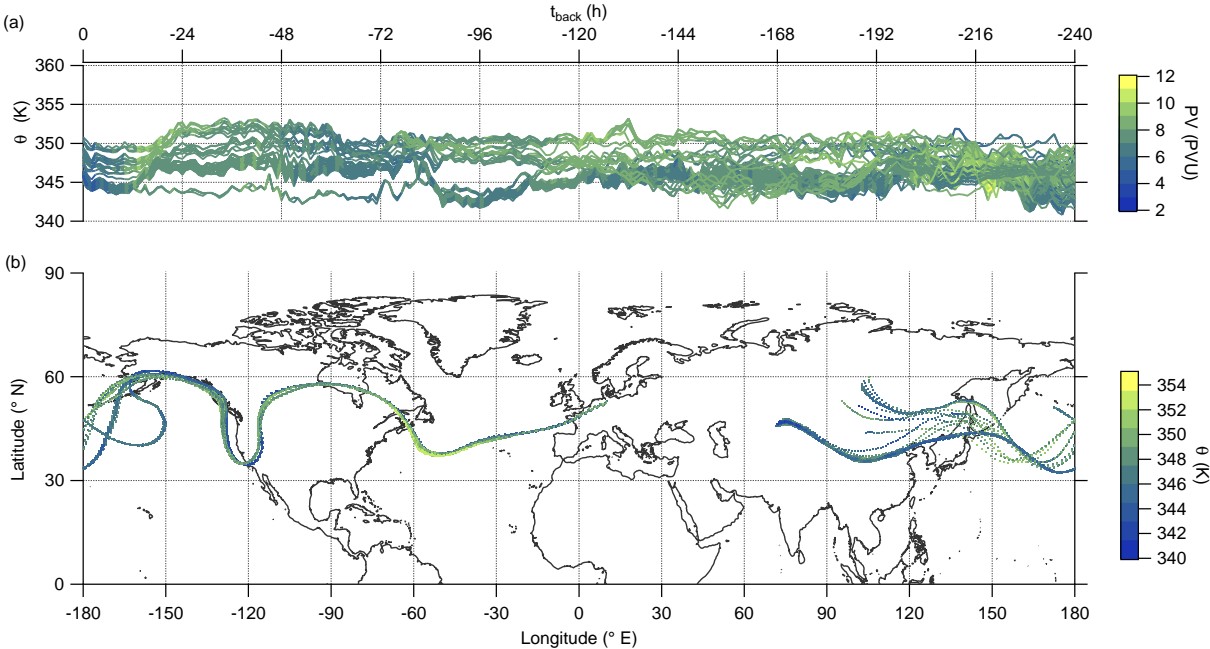

**Figure 10.** LAGRANTO 10 days back trajectories for cases with enhanced $SO_4^{2-}$ in the lowermost stratosphere. The upper panel (a) shows
a time series of $\theta$ color-coded with PV. The lower panel (b) shows the position of the trajectories on the map with $\theta$ as color-code.

Alternatively, the sulfate aerosol, as mentioned earlier, could originate from gas-to-particle conversion of $SO_2$ that was
mixed into the LMS. To examine this hypothesis, we start from Fig. 9, which shows that close to the tropopause, the $SO_2$
values are quite high and are decreasing along the mixing line whereas the aerosol total number concentration increases. This
is an indicator for ongoing gas-to-particle conversion in combination with cross-tropopause mixing. To confirm this possible
process it needs a strong source of $SO_2$ which is strong enough to transport the $SO_2$ in a short time into high altitudes with
as less as possible moist processes to not wash it out from the atmosphere. On possible source for such a process could
be volcanic activities. Therefore, we searched for volcanic eruptions in the period of two months before the measurements,
corresponding to the e-folding time of about 50 to 60 days (Jurkat et al., 2010). For the analysis, we used volcanic eruption
databases in combination with daily TROPOMI retrievals in different volcanically active regions to identify possible source
regions. Thereby, we identified the Kamchatka Peninsula in Russia as an origin of enhanced $SO_2$ emissions in the beginning of
April 2020. This corresponds to the archived reports by the Kamchatka Volcanic Eruption Response Team (KVERT) (Institute
of Volcanology and Seismology FEB RAS, 2023). In the report for 08 April 2020, an explosive eruption of the Sheveluch
volcano is described, with a volcanic cloud height reaching up to 10 km and thus into the tropopause region. We performed

HYSPLIT (Stein et al., 2015) forward dispersion simulations of the Sheveluch volcano plume and analyzed the eruption plume in the model in different heights to get a broad overview of the distribution. Therefore, we calculated forward trajectories in heights between 9000 and 11000 m with a vertical resolution of 500 m. In the following, we only consider the levels up to 9000 m (see Fig. D1) and between 10000 and 11000 m (see Fig. 11), because here the model shows differences in the plume and indicates possible mixing into the stratosphere.

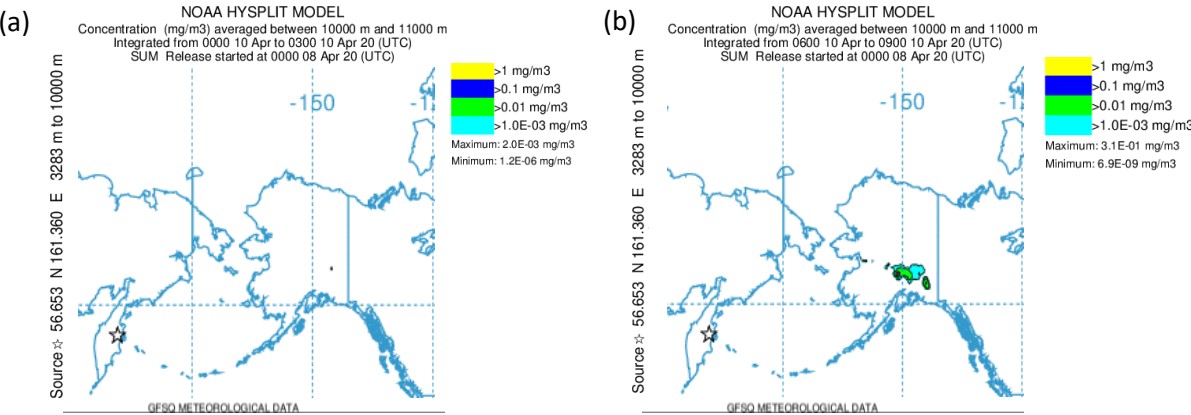

**Figure 11.** NOAA HYSPLIT dispersion model simulation (Stein et al., 2015) for the Sheveluch eruption on 08 April 2020 on basis of GFS meteorological data. The particle concentration between 10000 and 11000 m is depicted. (a) shows the averaging period on 10 April 2020 from 00 UTC until 03 UTC and (b) shows the next period on 10 April 2020 from 06 UTC until 09 UTC.

We observe a large volcanic plume up to 9000 m which is distributed, spread and stretched within the first three days over the Northern Pacific reaching over Alaska and also eastwards, close to the Hudson Bay, Canada (see Fig. D1). For the layer between 10000 and 11000 m, the results look completely different (see Fig. 11). Here, we observe no signal of the volcanic plume on the first day. However, more areas with volcanic plume influence occur over Alaska reaching towards Canada during the second day. These affected areas increase over time. Thus, also the HYSPLIT dispersion model results support the hypothesis of

mixing volcanic emissions into the stratosphere within the first three days. This is an additional indicator for the mixing of $SO_2$ into the stratosphere, resulting in higher $SO_2$ mixing ratios compared to the background. After the mixing of $SO_2$ into the LMS, the process of gas-to-particle conversion starts and forms particulate sulfate aerosol over several weeks. This results in the observation of higher mixing ratios of particulate sulfate in the LMS roughly seven weeks after the eruption, especially for research flight RF01 where the flight path was close to the jet stream and crossed one filament of volcanically influenced air

masses.

## 4 Conclusions

Usually trace gas correlations, such as $CO$-$O_3$ or $H_2O$-$O_3$, have been used to study mixing processes between the troposphere and stratosphere. In our study, we showed that in addition to trace gas measurements, also aerosol measurements, especially

particulate sulfate, can be applied to identify troposphere-stratosphere exchange. Furthermore, we showed that the method
to define the chemical tropopause proposed by Zahn et al. (2004) is in agreement with the dynamical tropopause definition
for our campaign and thus suitable for the separation of stratospheric and tropospheric air masses. Similar to the correlation
between CO and $O_3$, the correlation of $SO_4^{2-}$ and $O_3$ in the lowermost stratosphere also showed some variability induced by
mixing events. In a case study during the CAFE-EU/BLUESKY mission, we observed air masses with high sulfate mixing
ratios in the lowermost stratosphere, reaching values that are typically found at higher altitudes in the stratospheric aerosol
layer. Meteorological and dynamical parameters such as vertical wind shear and gradient Richardson number indicated that
mixing across the tropopause occurred in this region and along the transport and air mass history. Additionally, we found that
this anomaly of higher particulate sulfate in the lowermost stratosphere occurred during one single flight. During this flight,
we found one mixing line in the CO-$O_3$ correlation with increasing sulfate aerosol mixing ratios and total aerosol number
concentration towards the stratosphere. In addition, we used measurements of the quasi co-located DLR-Falcon aircraft. In
the same tracer-tracer correlation framework, using $HNO_3$ instead of $O_3$, we also found one mixing line. In contrast to the
increasing sulfate aerosol for the HALO measurements, we observed decreasing mixing ratios of $SO_2$ which is a precursor gas
for particulate sulfate. The combination of the sulfate aerosol mixing ratio, the total aerosol number concentration as well as the
reduction of $SO_2$ in the same measurement region led to the hypothesis of upward mixing of precursor gases and on-going gas-
to-particle conversion in the lowermost stratosphere. Here, we could identify volcanic activities on the Kamchatka Peninsula,
Russia, and the explosive eruption of the Sheveluch volcano as a possible source for the $SO_2$ in the tropopause region. The
Sheveluch eruption injected $SO_2$ directly into the upper troposphere from where it was mixed into the stratosphere, with
subsequent gas-to-particle conversion to sulfate aerosol. We can thus conclude that the chemical composition of the aerosol in
the lowermost stratosphere is affected by small-scale mixing processes, and that the ExTL can thus also be characterized by
aerosol properties. In addition to direct mixing of aerosol particles, the process of mixing of precursor gases with subsequent
gas-to-particle conversion needs also to be considered, as we showed in our case study. We intend to use this method in
the future with data obtained during previous airborne measurements in the UTLS to extend the analysis to a larger scale.
Furthermore, we aim to compare the results with chemistry climate model studies to see whether chemical transport models
can represent small-scale mixing across the tropopause and the associated gas-to-particle conversion processes. Another study
should investigate how models represent the influence of volcanic eruptions on the lowermost stratosphere.

*Data availability.* Data measured onboard HALO are available on request at the HALO database at https://halo-db.pa.op.dlr.de/mission/120.
The DLR-Falcon basic aircraft data and the AIMS data are available on request at the HALO database at https://halo-db.pa.op.dlr.de/mission/
119 (Re3data.Org, 2016) or on request to the authors. The trajectories are published on Zenodo (Kunkel and Joppe, 2024).

*Author contributions.* PJ set up the study together with JS, performed the data analysis and prepared the manuscript. KK and JS provided
the AMS and OPC data and supported the analysis. LT, AM and CV provided the $SO_2$ and $HNO_3$ data. HS provided CO and $O_3$ data

from DLR-Falcon. AZ provided the $O_3$ data from HALO. HF and LR provided CO data from HALO. HCL and DK provided model data and backward trajectories as well as the code for the cross sections along the flight path. All co-authors commented on the manuscript and discussed the presented results.

*Competing interests.*  The contact author has declared that none of the authors has any competing interests.

*Acknowledgements.*  PJ is funded by the Deutsche Forschungsgemeinschaft (DFG, German Research Foundation) – TRR 301 – Project-ID
428312742: "The tropopause region in a changing atmosphere", sub-project A04 coordinated by J. Schneider, S. Borrmann and F. Köll-ner. LT and CV are funded as well by the TRR 301 – Project-ID 428312742 in sub-project A01. The HALO measurement flights during CAFE-EU/BLUESKY were funded by the Max Planck Society. The authors thank the DLR team for making a campaign possible during the COVID-19 lockdown in Germany.
The authors thank NOAA Air Resources Laboratory for providing the HYSPLIT dispersion calculations on their website.

## Appendix A: Supporting information about the 2-D binned cross sections

In the following, we explain the adjustment of our bin scheme for the two dimensional binning analysis. Figure A1b shows the data distribution for the evenly distributed bin scheme in the vertical. Here, we see that many bins in the LMS contain less than 10 data points, so we could expect some bias in the median. Therefore, we enlarge the vertical bins at 350 K and above to 10 K to include more data points in one vertical bin and increase the statistic of the analysis without losing information.

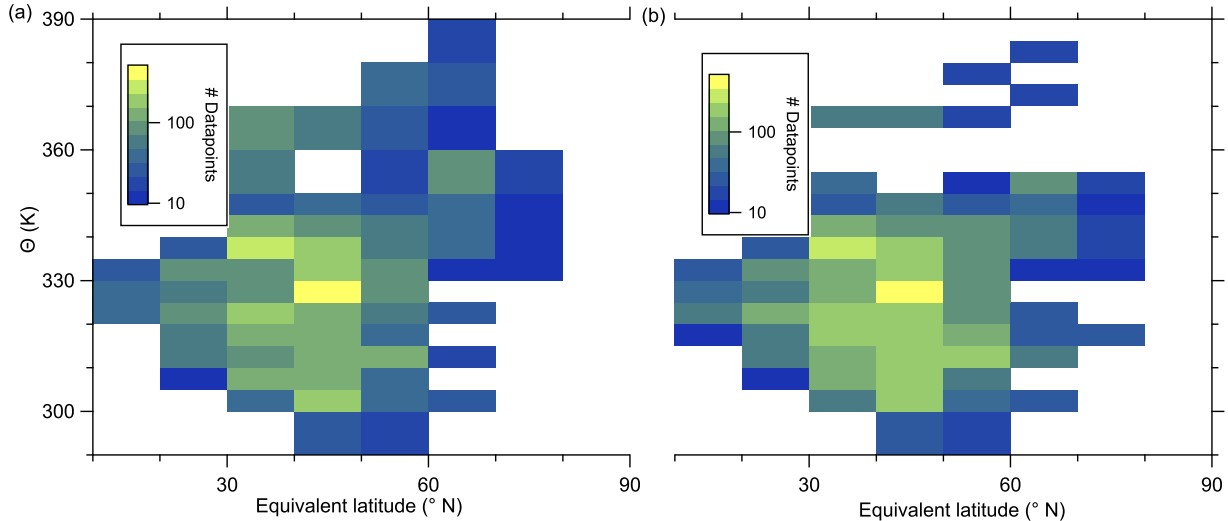

**Figure A1.** Number of data points in the 2-D cross sections (used in Fig. 5 and Fig. 6). (a) The used bin scheme with larger vertical bins starting at $\theta > 350$ K. (b) The evenly data distribution without any adjustment on the bin scheme.

## Appendix B:  Supporting information about the meteorological cross sections

This section offers some meteorological analysis for the flight segment with the observed particulate sulfate anomaly (see Fig. B1). Afterwards, we show some additional data along the back trajectories. More precisely, we show the time series of the Richardson number to indicate potential instable regions the air masses have crossed before the measurement (see Fig. B2).

First we want to introduce the used variables for our stability analysis, similar to e.g., Kaluza et al. (2019) or Kunkel et al. (2019):

$$N^2 = -\frac{g}{\rho_0}\frac{\partial \rho(z)}{\partial z} \quad \text{squared Brunt–Väisälä frequency} \tag{B1}$$

$$S^2 = (\frac{\partial u}{\partial z})^2 + (\frac{\partial v}{\partial z})^2 \quad \text{vertical wind shear} \tag{B2}$$

$$Ri = \frac{N^2}{S^2} \quad \text{gradient Richardson number} \tag{B3}$$

Turbulence may occur at a gradient Richardson number lower than 0.25

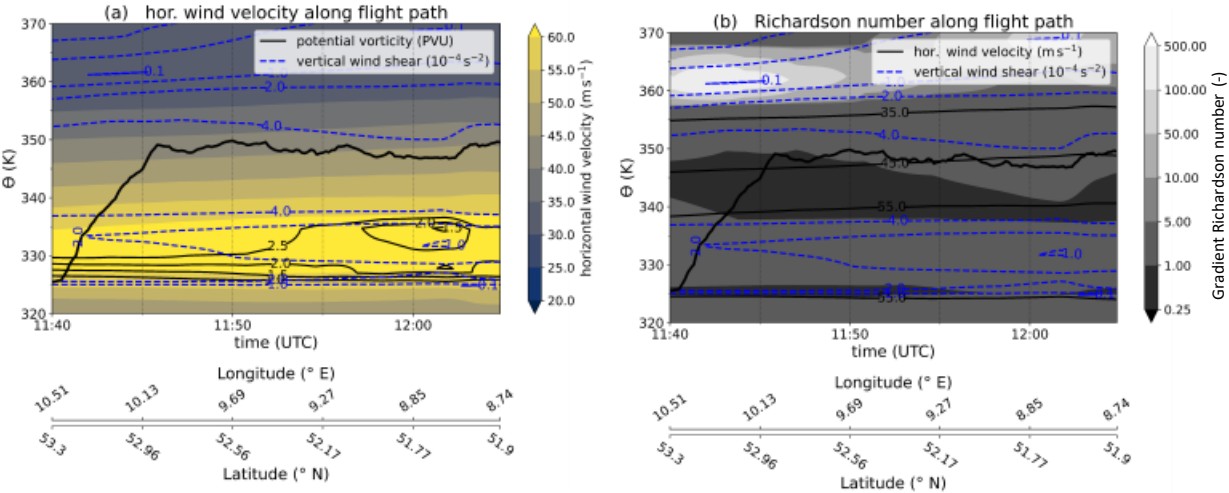

**Figure B1.** Meteorological cross sections along the flight path, calculated on the basis of ERA5-reanalysis data (Hersbach et al., 2020). (a) contains the horizontal wind speed as filled contour, the flight altitude as black bold solid line, the PV contour as thin black lines and the vertical wind shear as dashed blue contour lines. In (b) the filled contour changed to the gradient Richardson number and the thin black lines are contour lines of the horizontal wind speed. Latitude and longitude values are added to the time series for reference.

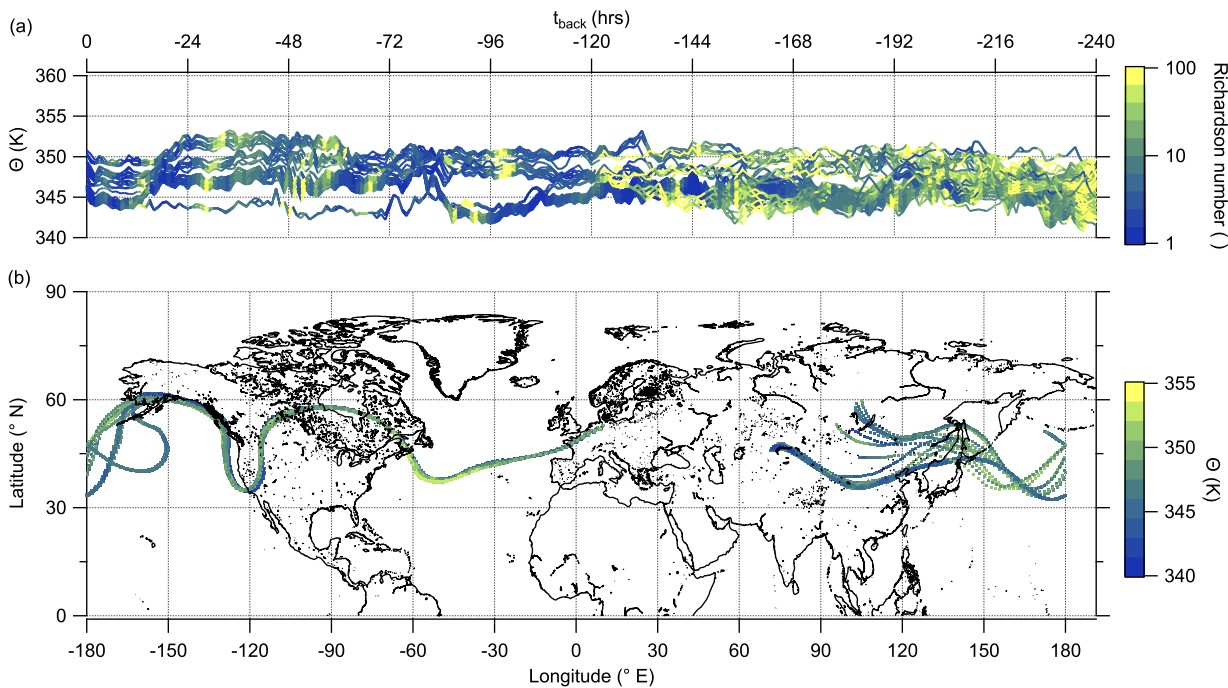

**Figure B2.** LAGRANTO 10 days back trajectories for cases with enhanced $SO_4^{2-}$ in the lowermost stratosphere starting in Northern Germany. (a) Time series of $\theta$ color-coded with the gradient Richardson number as marker for potentially instable regions along the trajectories. (b) The position of the trajectories on the map and the potential temperature as color-code.

## Appendix C: Supporting information about the sulfate anomaly

The following figures are supporting information on the observed anomaly of higher mixing ratios of particulate sulfate. This includes a detailed view on the research flight RF01 where the anomaly was observed (Fig. C1). Figure C2 locates the flight segment of the anomaly on the map including the quasi co-located DLR-Falcon flight path and the sulfur dioxide mixing ratio measured on the DLR-Falcon. The last figure (Fig. C3) highlights two selected research flights and their corresponding accuracy within the correlation of the complete data set.

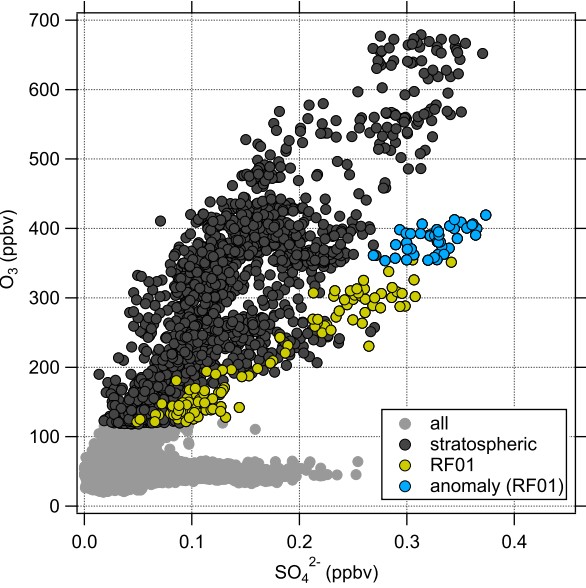

**Figure C1.** Scatter plot of the in-situ measured $SO_4^{2-}$ and $O_3$ containing/including the whole campaign data in bright grey. Stratospheric data points are colored in dark grey. The mixing event of research flight RF01 is indicated in yellow and the sulfate anomaly identified during this flight in blue.

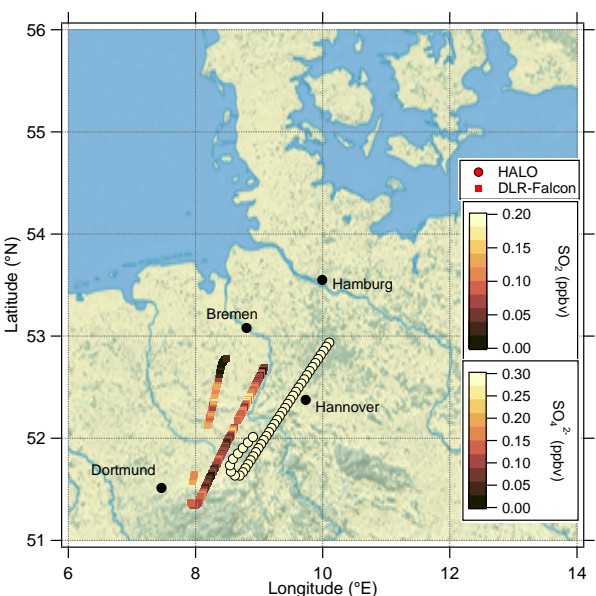

**Figure C2.** Location of the analyzed flight segment during the research flight on 23 May 2020 between 11:45 and 12:05 UTC where the mixing event in the vicinity of the jetstream was observed. The flight path of HALO are the filled circles with color-coded with the sulfate mixing ratio and the flight path of DLR-Falcon are the filled squares color-coded with the $SO_2$ mixing ratio. The map was created from public domain GIS data found on the Natural Earth website (http://www.naturalearthdata.com; last access: 02 January 2024)

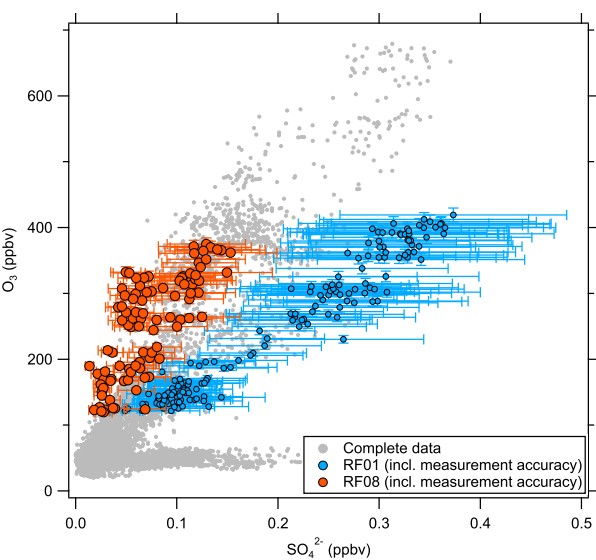

**Figure C3.** Scatterplot of particulate sulfate against $O_3$ for the complete campaign data set. The data points for flights RF01 and RF08 are highlighted together with their measurement accuracy as examples.

The following maps (Fig. C4) of the measured stratospheric air during RF01 help to interpret both mixing lines found in Fig. 8. Herewith, we can differ between the measured elevated particulate sulfate over Northern Germany and more subtropical air over Southern Germany with respect to their characteristics.

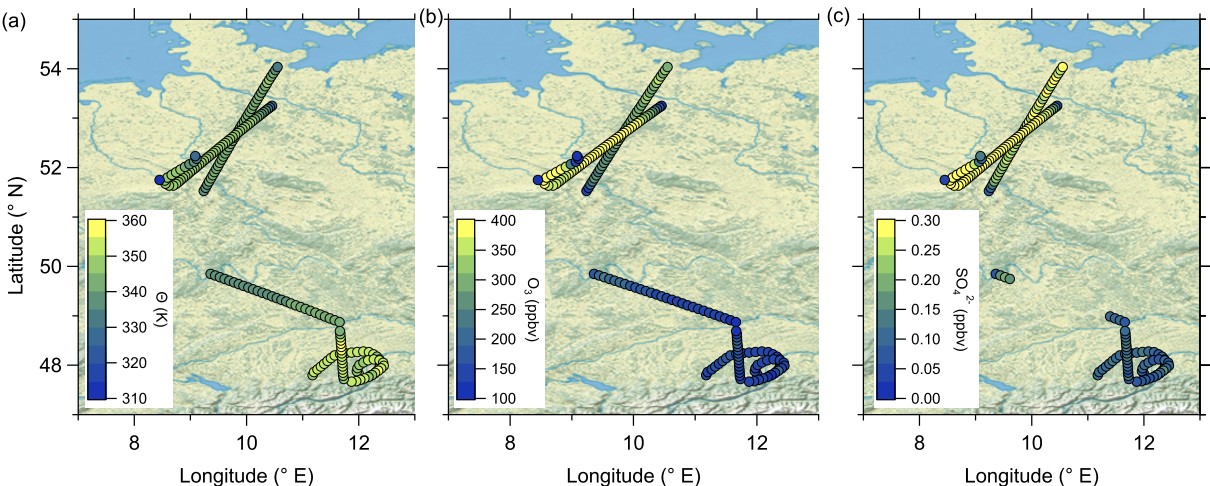

**Figure C4.** The stratospheric flight segments of RF01 on 23 May 2020 are shown and color-coded with (a) potential temperature $\theta$, (b) $O_3$ mixing ratios and (c) particulate sulfate mixing ratios.

**395 Appendix D: Supporting information about the HYSPLIT dispersion simulation**

The figure shown in this section is in addition to Fig. 11 and shows the similar variables but for the altitude range from sea level up to 9000 m to show the entire volcanic main plume and its distribution.

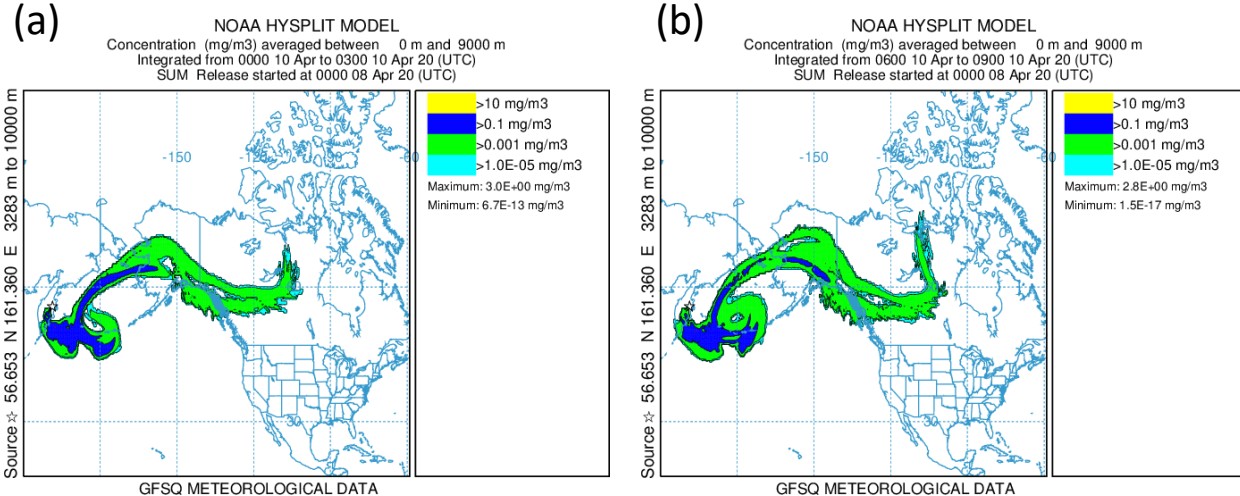

**Figure D1.** NOAA HYSPLIT dispersion model simulation for the Sheveluch eruption on 08 April 2020 on basis of GFS meteorological data (Stein et al., 2015). The location of Sheveluch volcano is given by a little star. Further, the particle concentration averaged between sea level and 9000 m is shown. (a) includes the averaging period on 10 April 2020 from 00 UTC to 03 UTC and (b) shows the next period on the same day from 06 UTC to 09 UTC.

## Appendix E: Additional measurement data conducted by the C-ToF-AMS

This section gives an overview on the complete dataset produced by the C-ToF-AMS in combination with the OPC that is integrated into the instrument system. In the following, we show vertical profiles of the aerosol species measured by the AMS relative to the potential temperature and the geometric altitude to support the anomalous observation of the sulfate concentration described in the case-study.

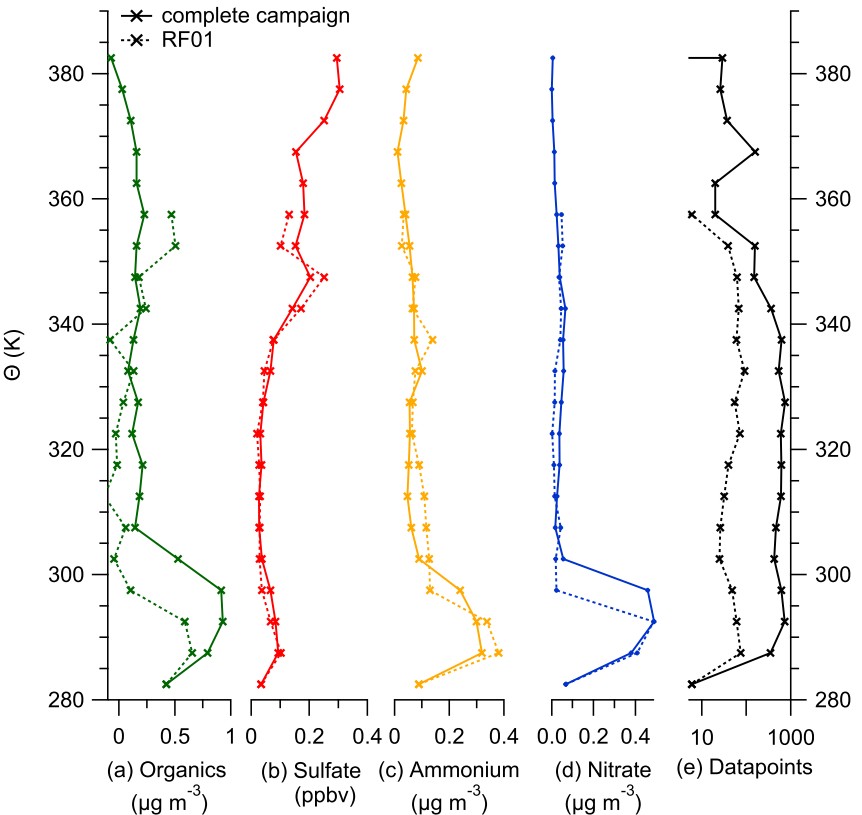

**Figure E1.** Binned vertical profiles for five K potential temperature bins for all species measured by the C-ToF-AMS: (a) organic aerosol, (b) sulfate aerosol, (c) ammonium aerosol and (d) nitrate aerosol. The number of datapoints for each bin is shown in (e). The vertical profiles are divided into the complete dataset (solid lines) and data only from the case-study RF01 (dotted lines).

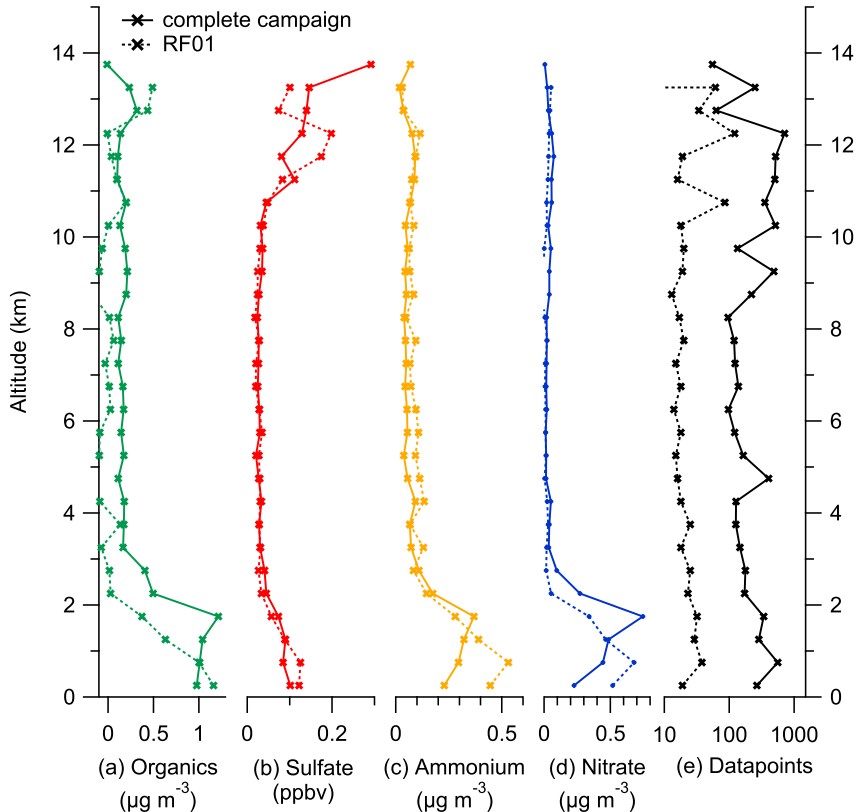

**Figure E2.** Binned vertical profiles for 500 m altitude bins for all species measured by the C-ToF-AMS: (a) organic aerosol, (b) sulfate aerosol, (c) ammonium aerosol and (d) nitrate aerosol. The number of datapoints for each bin is shown in (e). The vertical profiles are divided into the complete dataset (solid lines) and data only from the case-study RF01 (dotted lines).

The following plot shows the time series of the altitude and the aerosol measurements conducted by the C-ToF-AMS during RF01. The orange box indicates the time period where the sulfate anomaly was observed. Further, the different chemical composition of the probed air masses is visible, similar to the described characteristics in Fig. C4 and Fig. 7.

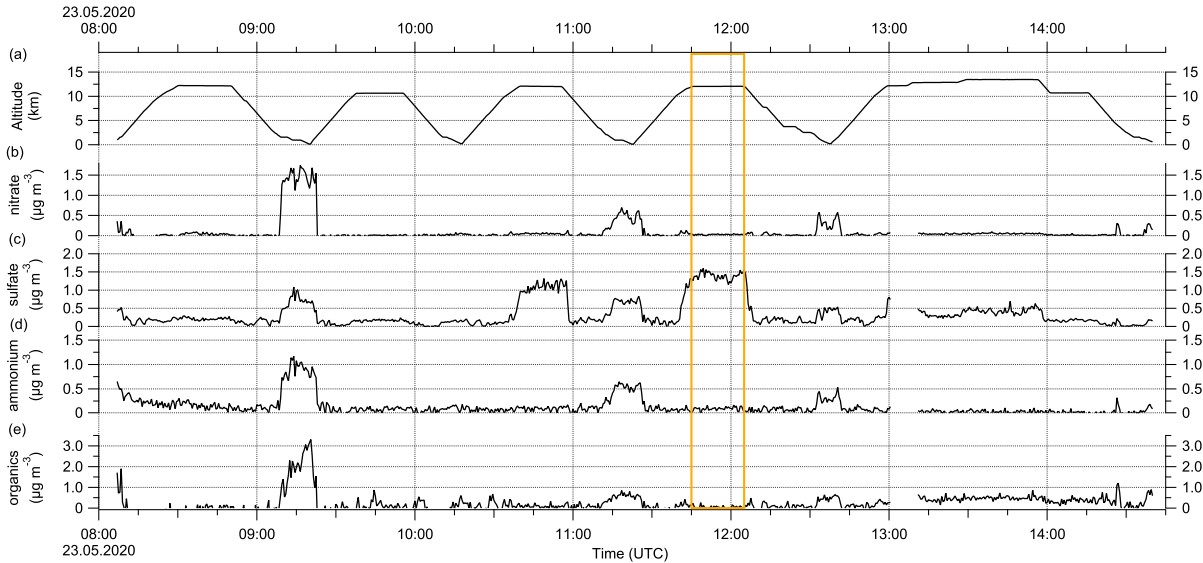

**Figure E3.** Time series of the aerosol measurements during RF01 on 23 May 2020 in 30 s time steps. (a) Altitude, (b) nitrate aerosol, (c) sulfate aerosol, (d) ammonium aerosol and (e) organic aerosol mass concentration. The orange box marks the period with the observed sulfate anomaly described in the text and shown in Fig. C1.

In addition to the chemical composition analysis, we looked at the aerosol size distribution and number concentration measured by the OPC. Therefore, we compare different stratospheric measurement periods. In Fig. E4, we show the size distribution corresponding to the 20 min of the sulfate anomaly and compare it with 20 min measurements for the stratospheric part over Southern Germany under background conditions. During the anomaly, we observe up to two times more particles within the individual size bins below one μm.

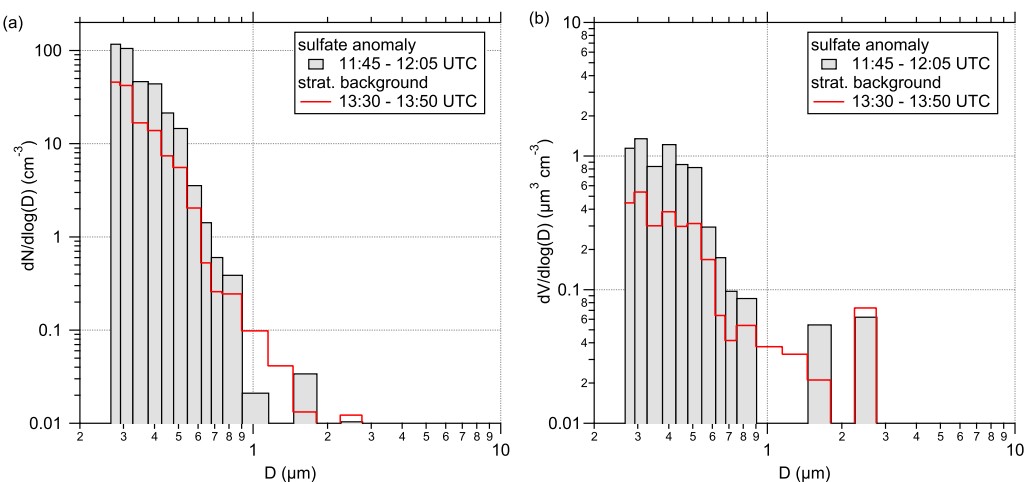

**Figure E4.** Size (a) and volume distribution (b), STP corrected, measured by the OPC. The distributions are averaged over a 20 min time period for two different measurement regimes. One is the described sulfate anomaly (grey-filled bars) and the other regime is a time period of stratospheric background (red lines) later during this flight as comparison.

To illustrate the chemical composition of the formed particles in Fig. 9a, we correlate the total number concentration N with the individual aerosol species in the next figure.

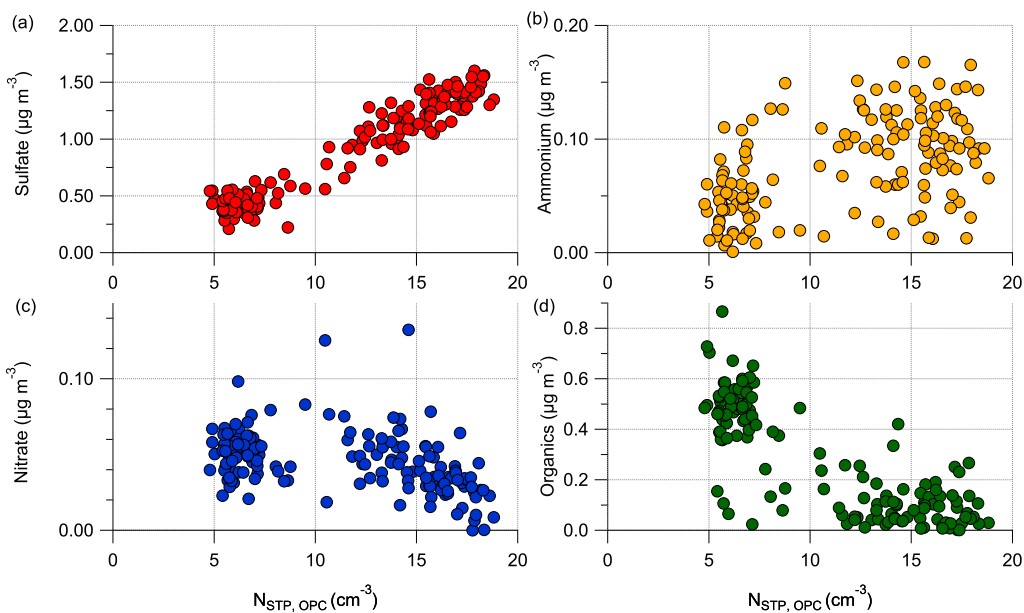

**Figure E5.** Correlation of the total aerosol number concentraion N, STP corrected, measured by the OPC with the aerosol species measured by the C-ToF-AMS: (a) with particulate sulfate, (b) with ammonium, (c) with nitrate and (d) with organic aerosol. The shown data represents the stratospheric measurements during RF01.

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
