# Peer review of "The influence of extratropical cross-tropopause mixing on the correlation between ozone and sulfate aerosol in the lowermost stratosphere"

_EGUsphere, 2024_

## Author Comment (AC1)

**EGUsphere-2024-7**

The influence of extratropical cross-tropopause mixing on the correlation between ozone and sulfate aerosol in the lowermost stratosphere

P. Joppe et al.

**Author comments to Reviewer #1**

*The reviewer comments are written in this font style and color.*

Our answers are written in this font style and color.

Changes in the revised version of the manuscript are written in red.

*In this work the authors suggest an alternative tracer-tracer metric for exploring the 'Extra-tropical Transition Layer' using the correlation between sulfate aerosol and ozone. The work is based on aircraft measurements of aerosol sulfate using an Aerosol Mass Spectrometer aboard the DLR HALO aircraft and trace gas measurements from the HALO and DLR-Falcon from 2020 over central Europe. In part 1 the authors show a robust relationship between sulfate and ozone near the tropopause, and seek to use the variability in this relationship to identify atmospheric processes, of particular interest is the persistent minima in sulfate mixing ratio near the tropopause. In part 2, the authors identify a specific case of enhanced sulfate from one ascent in one flight and then perform an analysis to conclude that this enhancement is the result of mixing through the ExTL followed by gas to sulphate aerosol conversion from a distant volcanic eruption.*

We thank the reviewer for reading our paper carefully and providing comments to improve the manuscript.

*Major comments:*

1. *The correlation between aerosol sulfate and ozone is interesting, and establishing a background ratio is useful to identify perturbations in sulfate aerosol on top of the steeply sloped vertical aerosol gradient. However, the analysis of aerosol sulfate seems to be performed in isolation from the other aerosol properties that must surely have been made on the same measurement platform, or even by the same instrument (the C-ToF-AMS). In terms of the analysis in part 2 off the enhanced sulfate plume observed in RF01, a discussion of the aerosol size distribution and the variation of other chemical constituents (particularly organics and nitrates) would provide both context and a much more convincing argument that this particular enhancement is volcanic in origin. Furthermore, including the size distribution of the observed sulfate would reassure the reader the analysis of the C-Tof-AMS is actually capturing the bulk of the sulfate aerosol within the somewhat limited size range.*

It may seem that the performed analysis was done isolated from the other properties that were measured by the C-ToF-AMS (e.g., organics, nitrate and ammonium mass concentration as well as the particle size distribution measured by the SkyOPC), but we also looked carefully at the whole set of available data. The results are not shown here explicitly, because we did not observe the same unexpected results in the data as we did for the sulfate concentration. However, we added the time series and vertical profiles of all species measured by the C-ToF-AMS as well as the size distribution measured by the OPC to the supplement information (Fig, E1- E4) to complete the picture.

2. *The authors make a convincing argument that the enhancement observed on RF01 is in fact anomalous and possibly of tropospheric origin (figures 5, 6 and 8). However, the argument that this enhancement is of volcanic origin and is due to gas to particle conversion in the ExTL is less convincing.   The authors seem to start with the conclusion that this is in fact a volcanic event, then search for a candidate volcano and use a combination of forward and backward trajectory models to try and link the observation to the event, which is convoluted and unconvincing.   A more convincing argument to show this event is volcanic in origin (and not anthropogenic for example) would be to present the total aerosol composition and size distribution, as discussed above.   It is also interesting that RF01 only observed this enhancement on one or maybe two climbs of the flight, despite reaching higher altitude and theta on the subsequent climb.   Is this enhancement limited in geographic scope?   Without a broader analysis and a more convincing identification of the source of gas phase sulfur, part 2 of the paper is of less scientific relevance than part 1.*

We agree with the reviewer that Figs. 5, 6 and 8 are the main figures which show the anomalous observation of enhanced sulfate in the lowermost stratosphere. We want to point out that Fig. 9 is also (very) important for the argument of gas-to-particle conversion. It clearly shows enhanced sulfur dioxide in the UTLS region (as described by Tomsche et al. (2022)) which decreased along one mixing line in the correlation framework. To emphasize the volcanic origin, we added the size distribution and the vertical profiles of the complete AMS species to the supplement as mentioned above. These show that only the particulate sulfate is enhanced within the anomaly. Furthermore, this enhancement of sulfate aerosols correlates with an increase in the total aerosol number concentration along the observed mixing line which shows that no tropospheric compounds are directly mixed into the LMS. Additionally, the size distribution shows that during the measurement period of the anomaly up to a factor of two more particles with diameters below one µm were observed. Further, we want to state that models show that the Shiveluch volcano had an impact on the measurement region, but these results are preliminary and planned to be published later this year.

Regarding the geographic scope of the anomaly, we have to admit that the flight planning during the campaign was focused on the PBL and the troposphere and not on the UTLS region, so the results might be biased by the flight pattern with only short times in the UTLS region. In addition, in the planned publication mentioned above, we will analyze the geographic extent of the volcanic plume in model simulations.

*Minor Comments:*

*The figures are not well aligned with the text that references them, making it difficult to reference while reading the text.*

We reorganized the figures in the revised version with fixed positions to simplify reading and referencing.

*Line 28: I am not sure what you mean by 'photochemical dissolving'?*

We rephrased the sentence to describe the origin of OCS as follows:

OCS is the main sulfur containing trace gas in the atmosphere with direct emissions by the oceans or biomass burning as well as photochemical production by oceanic emissions like DMS or CS2.

*Line 71 / Section 2.1: It may be important to note the limitation of the HALO aircraft measurement with respect to altitude – it appears to be limited to a peak altitude of around 12km, which may not really reach the top of the ExTL in May at latitudes below about 45N leading to a bias to higher latitude measurements.*

We agree with the reviewer that it is important to note the limitation of covering the complete vertical structure of the ExTL in May. However, peak altitudes, also during RF01, were around 14 km. HALO's ceiling altitude is above 15 km, but during this campaign the scientific questions focused on vertical profiles close to urban areas and the tropospheric chemistry during COVID-19 lockdown. We added the following sentences:

… This point leads to flight planning during the campaign with focus on urban areas and low altitude profiles and less on studying processes in the UTLS region. Therefore, it was not possible to conduct measurements over the complete vertical extent of the ExTL during May 2020. Nevertheless, we were able to obtain measurement data up to 14 km altitude. During the campaign period…

*Line 163 and Figure 3: The persistent minima in sulfate at the 'ozone tropopause' of 90 – 120 ppb is surprisingly consistent and robust. This may be worth more than a passing mention.*

We agree with the reviewer that this observation very robust for the dataset, but our current status of research does not allow any conclusion about a climatology of particulate sulfate at the chemical ozone tropopause, or especially one about climatology of the sulfate-ozone correlation in the LMS. We added the following sentence:

This observation of low particulate sulfate aerosol amounts at the chemical tropopause is very robust over the whole campaign period, such that it might be controlled by atmospheric processes that need more investigation.

*Line 186: It is not clear what you mean by 'not connected to the stratosphere' when the airmass meets both the PV > 2 and ozone > 120 ppb criteria by a large margin.*

The connection to the stratosphere in this case refers to a connection to the bins with a potential temperature above 370 K and sulfate mixing ratios around 0.3 ppbv which represent the lower boundary of the stratospheric aerosol layer.

We reformulated the sentence as follows:

The observed sulfate anomaly occurs in Fig. 5b between 40◦ N and 45◦ N at potential temperatures between 345 K and 350 K and is not connected to the observed stratospheric aerosol layer that starts at higher altitudes, above the 370 K isentrope (see Fig. 5).

*Line 219:  Won't dilution equally impact all tracers and not just CO?*

That is correct. In contrast, sulfate is formed during its transport into the stratosphere, resulting in an increasing ratio of sulfate to other tracers such as CO. We reformulated:

In contrast to sulfate, CO will decrease during the transport into the stratosphere, both by dilution and photochemical destruction, with an atmospheric lifetime of one to three months (Seinfeld and Pandis, 2016).

*Line 225:  The stratospheric water vapor background is closer to 5 ppm, why would you expect it to be 10 – 15 ppm, and doesn't Fig 8e show that the water vapor us is in the range of 10 – 20 ppm?*

We agree with the reviewer that the stratospheric water vapor background is around 5 ppm and it was misleadingly written. We reformulated the line to the following:

If the air masses were stratospheric origin, we would expect O3 mixing ratios higher than 400 ppbv and a water vapor mixing ratio around 5 ppmv. Instead, we observe lower O3 mixing ratios and water vapor mixing ratios around 10 – 20 ppmv.

*Figure 7: Is there an easy way to show latitude on this figure?  It may help understand the geographic extent of the anomalous layer and why it is only observed on some of the climbs?*

We added the timeseries of the latitude to Fig. 7 to compare the same measurement regions easily. The anomaly with the highest mixing ratios of particulate sulfate is observed on the highest level around 12:00 UTC. Moreover, we observe the same air mass characteristics with weaker signals between 10:30 and 11:00 UTC within the same measurement region. In contrast to these two observations over Northern Germany, we measured another air mass in the same altitude with a different composition over Southern Germany between 13:00 and 14:00 UTC.

*Figure 8: The caption does not agree with the figure labels, and the color bar labels for theta are incorrect.*

We are sorry for this mistake and corrected the figure including the caption.

*Figure A1: The y-axis label is incorrect.*

We are sorry for this mistake and corrected the figure (y-axis label).

---

## Author Comment (AC2)

**EGUsphere-2024-7**

The influence of extratropical cross-tropopause mixing on the correlation between ozone and sulfate aerosol in the lowermost stratosphere

P. Joppe et al.

**Author comments to Reviewer #2**

*The reviewer comments are written in this font style and color.*

Our answers are written in this font style and color.

Changes in the revised version of the manuscript are written in red.

*Any field observations and measurements are valuable for model development. Global aerosol models predict that certain amounts of sulfate will accumulate at high altitudes above 12 kilometers, and this observation confirms this. This is an observational paper reporting aircraft field measurements during the CAFE-EU/BLUESKY mission. The authors try to find out the reasons behind these observations. Therefore, the objectivity of the analysis is very important for this article. The authors further analyzed the observations using a variety of methods and combined all analyzes in an attempt to trace the origin of the entire observation. Overall, the material in this manuscript is well organized and well written, and therefore deserves acceptance and publication.*

*This manuscript still leaves some room for improvement. Due to the limitations of observation, the conclusion drawn from the current limited observations is only a possibility or a reasonable explanation. In addition, the analysis in this manuscript focuses on the sulfate anomaly discovered on Flight 01. There are approximately 45 observation points, and its proportion in all observations is relatively small. Other observations with similar meteorological conditions did not show similar characteristics. This often means that reality is more complex, and there may be something behind it that we currently don't know about. Therefore, I hope that the authors will be aware of this in order to provide more explanations and treat the conclusions with more caution in a revised version of this manuscript.*

We thank the reviewer for the careful reading and the helpful comments to improve our manuscript. We appreciate the recommendation for publication of our work.

*The following are specific comments:*

1. *Line 8, "non-refractory aerosol --- to 800nm", Can I assume that this category of sulfate roughly represents the total amount of it in the stratosphere?*

   Regarding to previous studies in the literature and in combination with our measurements of the particle size distribution by the OPC, the aerosol measured by the C-ToF-AMS represents the

typical amount of sulfate aerosol. For example, Tilmes and Mills et al. (2014) found the typical effective radius of sulfate aerosol at 170 nm.

Line 24: Sulfate aerosol [...] and has an average radius in undisturbed conditions of 170 nm (e.g., Tilmes and Mills, 2014)

2.2 Instrumentation

[...] Integrated into the C-ToF-AMS, we use an optical particle counter (OPC) manufactured by GRIMM (SkyOPC 1.129) to measure the aerosol size distribution in 31 size channels from 250 nm to larger than 32 µm.

2. *Line 10, "background sulfate", Please add discussion about this part*

Here we wanted to express that the slope of the correlation exhibits some variability over time with respect to the average slope observed during the campaign period. We reformulated the text to make this clear:

The correlation exhibits some variability exceeding the mean sulfate to ozone correlation over the measurement period.

3. *Line 14, "volcanic SO2", Judging by the text, the analysis supporting this argument is insufficient.*

We hope that our revised version, with extended data and additional analysis, will support this argument for volcanic SO2 better and make the evidence of volcanic influence clearer.

4. *Line 90, "the accuracy of the AMS is about 30%", Perhaps a discussion of how 30% accuracy affects the results of the analysis could be added to the text.*

The accuracy of 30 % of the AMS is caused by the ionization efficiencies, the inlet transmission efficiency and the collection efficiency (see auxiliary material to Bahreini et al., 2009). These values will not change over the course of a two-week field campaign. Thus, the difference between the slopes observed on different flights during the campaign is not affected by the accuracy.

Please see also Fig. C3 in the appendix, where the 30 % accuracy is added to the measured values. We added a statement on the accuracy to the revised version of the manuscript:

Note that the accuracy of the C-ToF-AMS of about 30 % (Bahreini et al., 2009) does not affect the observed different slope regimes in the correlation of sulfate aerosol and ozone, because the quantities determining the accuracy (ionization efficiency, collection efficiency and inlet transmission efficiency) do not change over the short period of a two-week measurement campaign.

5.  *Line 96 to 99, "O3 and HALO ---, respectively", Is CO measured on the HALO?*

Yes, CO is measured on both aircraft. On HALO, it is measured by the TRISTAR instrument (Line 95) and on DLR-Falcon, CO is measured by a cavity ring-down spectrometer (Line 98).

6.  *Line 102, "on gaseous SO2", Is SO2 only measured on DLR-Falcon?*

Yes, unfortunately SO2 was only measured on DLR-Falcon.

7.  *Line 105, "the total uncertainty --- for HNO3", uncertainty is a bit high*

The total uncertainty for HNO3 of 16 % is in line with earlier measurements performed with the same instrument. It was not possible to improve this value for the data in this work due to the ad-hoc nature of this specific instrument deployment in the BLUESKY flight campaign.
We changed the reference for this value from the PhD thesis (Marsing, 2021) to a more recent publication (Ziereis et al., 2022).

8.  *Line 109 to 110, "there hours --- in the horizontal", 3 hours compared to 30 seconds; 50 kilometers compared to 6 kilometers.*

We agree with the reviewer that the scales are different, but this method allows us to compare model reanalysis data with airborne measurements to get some additional meteorological information which otherwise would not be possible. This method of interpolating model data to the flight path has been used and evaluated multiple times in previous studies (e.g., Lachnitt et al., 2023).

9.  *Line 111, "potential vorticity and equivalent latitude", Please briefly explain how equivalent latitude is calculated; Is it based on model predictions or measurements? If it is based on forecasts, how does the resolution of the ERA5 data affect the analysis?*

The calculations are performed over isentropic surfaces from ERA5-reanalysis data (240 - 2000 K) with tropospheric vertical spacing of 10 K and in the measurement region and the stratosphere 5 K. Potential vorticity and equivalent latitude are calculated on isentropic surfaces from the model (reanalysis) data interpolated on potential temperature. This explanation is also added to the revised version of the manuscript.

The equivalent latitude is a framework to get information of the origin of a measured air parcel over the potential vorticity. Therefore, a contour line having the same potential vorticity and potential temperature is transformed into a pole centered circle. The equivalent latitude is the enclosing latitude of this circle. It can be used to consider reversible adiabatic transport by e.g., planetary waves (e.g., Lary et al., 1995; Hegglin et al., 2006; Krause et al., 2018) due to the properties of potential vorticity (conservation under adiabatic processes). These calculations are done over isentropic surfaces from 240 up to 2000 K from the ERA5 reanalysis data interpolated on potential temperature.

*10. Line 117, "231", why 231? please explain*

The statistics for the trajectory analysis can be increased by a large number of trajectories. Therefore, we decided to start additional trajectories together with the ones directly at the flight path. We chose one possibility of a grid structure to start trajectories in a small area around the location of the aircraft. The method is described in lines 116 and 117. By summing up all these starting points, we get the number of 231 start locations for each release time along the flight path. We rephrased the text to make the number clearer.

Therefore, we initialize a set of 231 trajectories every 30 seconds along the flight path. The starting points of each trajectory set are placed in a three-dimensional cross around the initial point of the flight path, to gain a better statistic and to minimize interpolation errors between the measurements and the model grid. More specifically, we take the location of the aircraft and add five additional points every 0.01 degree in all four horizontal directions (north, east, south and west) resulting in 21 points arranged in a cross shape (including the aircraft position/location). This cross pattern of 21 points is repeated in 10 additional vertical levels in one hPa steps, five levels above and five levels below the flight altitude. Thus, we get a total of 231 trajectories starting locations at each release time, …

*11. Line 137, "two modes", what are they?*

We added the following explanation to the revised manuscript version, to simplify reading of this paragraph:

The first mode is resulting from air freshly mixed into the stratosphere (from the troposphere) with PV values close to the dynamical tropopause. The second mode with values larger than 8 PVU describes the deep stratospheric branch of the Brewer-Dobson-circulation with air originating from the high stratosphere with no tropospheric influence.

*12. Line 138, "can be --- air masses", reference*

We added a suitable reference for the connection of PV and stratospheric age to the revised version of the manuscript:

Bönisch et al., ACP, 2009 (https://doi.org/10.5194/acp-9-5905-2009)

*13. Line 146, "observations", Does it contain DLR-Falcon data?*

Figure 3 only included measurements performed onboard HALO, so here the DLR-Falcon is not involved in the analysis. DLR-Falcon data are shown in Fig. 1 (flight path), Fig. 9 (SO2, CO and HNO3) and Fig. C2 (flight path and SO2).

The observations made on HALO during the CAFE-EU/BLUESKY campaign confirm this (Fig. 3).

*14. Line 147, "Fig. 3", If my observations are correct, the lowest sulfate concentration observed in each flight measurement in the stratosphere appears to be different. If true, does the author have any explanation?*

We agree with this observation by the reviewer that the lowest sulfate concentration of each measurement flight is different. One possible explanation is that sulfate aerosol is partly dependent on local sources in the troposphere, such that we expect higher values close to industrial areas. Another explanation might be the method we use to determine the tropopause. For this, we use the chemical ozone tropopause based on daily threshold values. These thresholds underly a seasonal cycle, so there might be a coupling to tropopause height (in geometric coordinates), and by this also to the sulfate aerosol concentration.

*15. Line 150, "900 to --- measurement flights", Curious to know what the author has to say about why each flight observation has its own unique ozone to sulfate ratio*

We agree with the reviewer that this observation is of high interest. We are currently working on a collection of all airborne data we have sampled in different projects and measurement campaigns to obtain a climatology on the ozone-to-sulfate ratio. Thus, at the current stage we cannot answer this question completely. One possible explanation might be that the different slopes depend on the different flight planning and scientific goals of the research flights, and on the different meteorological conditions. We expect that this variability decreases when analyzing data measured on less specific flight routes and obtained over a longer time period, such as for example the IAGOS-CARIBIC project.

*16. Line 163 to 164, "Figure 3 --- chemical tropopause", What is the background concentration of sulfate in the chemical tropopause?*

We did not intend to relate our observation to a background sulfate concentration at the chemical tropopause. We could not find literature on this topic, and our dataset presented here is too specific and too short on timescales to define a background concentration of sulfate aerosol at the chemical tropopause. This is a future project as described in our answer to comment 15. With these sentences we wanted to state that there is no direct mixing of high concentrations from the troposphere to the stratosphere. We reformulated this sentence to make it clear:

The low sulfate mixing ratios at the chemical tropopause (Fig. 3) show that direct mixing of high sulfate aerosol concentrations from the troposphere to the stratosphere was not observed during the campaign, so some other processes need to be taken into account.

*17. Line 175 to 176, "The presence --- mixing processes", why is stratospheric CO mixing line set to less than 20ppbv?*

This line is meant to be a visual indicator of the mean values of the pure stratospheric and tropospheric regimes in this scatter plots. We agree that the stratospheric regime usually starts at higher CO mixing ratios (at approx. 20 ppbv) than shown here by this line. This is also written in the caption of Fig. 4.

18. *Line 179 to 180, "However, --- see also Fig. C1)", It seems that the entire RF01 flight data is abnormal, not just the part where the sulfate concentration is above 0.3 ppbv. Because the linear regressions for ozone and sulfate appear to match very well across observations.*

We agree with the reviewer that the entire RF01 is influenced by higher sulfate concentrations in the lowermost stratosphere. Here, we wanted to focus on the case with very high sulfate concentrations at the observed ozone mixing ratio of about 400 ppbv. In our answers to Comment 27 and 30 we will give one more specific reason for this anomalous observation with respect to the measurement region and possible air mass origin.

Figure C4 added to the appendix of the revised version of the manuscript. These maps of the measured stratospheric air during RF01 help to interpret both mixing lines found in Fig. 8. Herewith, we can differ between the measured elevated particulate sulfate over Northern Germany and more subtropical air over Southern Germany with respect to their characteristics.

19. *Line 185, "Fig. 5", Are the potential temperatures shown in Figure 5 measured? if so, have you compared the potential temperature of the measurements with the potential temperature of the ERA5 data set? Since the equivalent latitude is from ERA5.*

Figure 5 includes the potential temperature derived from the measured temperature and pressure at the aircraft but not the model data to compare the atmospheric conditions in the measurement region.  The comparison of the HALO measurements and the interpolated model data show a good agreement with a median deviation of less than 0.2 K. The comparison between the measured Falcon data and the model data shows that ERA5 underestimates the potential temperature around 1 K.

20. *Line 197, "around 1", >0.25*

We are aware of the critical threshold of 0.25 and reformulated the sentence to show that the Richardson number is close to the critical threshold for the occurrence of turbulence.

This also results in a reduction of the gradient Richardson number in the same bin to values close to the critical threshold of 0.25 (see Fig. 6c) …

*21. Line 198, "the static --- stratospheric values", In addition to the region focused on in this study, many other regions also show the potential for air masses to transition from the troposphere to the stratosphere*

In this line, we describe that the observed static stability does not show an anomalous behavior in contrast to the vertical wind shear. However, we agree with the reviewer that from the stability analysis there are more regions which show the potential for cross-tropopause mixing. We chose the region for the case-study because there we can observe the strongest signal for potential cross-tropopause mixing.

However, regarding the stability analysis we probed several regions which show favorable conditions for instability and cross-tropopause mixing. Nevertheless, here we focus on the region with the strongest signal, where the observed sulfate anomaly was measured.

*22. Line 211, "five ppmv", approximately 10 ppmv in Figure 8*

The 5 ppmv refer to the stratospheric background according to Hegglin et al. (2009). We agree with the reviewer that the measured mixing ratio is approximately 10 ppmv and thus slightly higher than the stratospheric background.

We rephrased:

The H2O-O3 method follows the same principle, with high water vapor mixing ratios in the troposphere and a constant stratospheric background value of around five ppmv (Hegglin et al. (2009). In our data set, the lowest observed H2O values are around 10 ppmv, indicating that we did not fully reach stratospheric background conditions.

*23. Line 212 to 213, "all of --- mixing lines", please elaborate*

In Fig. 8, we observe two different slopes of mixing lines. We observe one mixing line with a steeper slope which in addition shows the described anomaly on the upper end. The other observed mixing line has a shallower slope and is bounded at the upper edge at 200 ppbv O3.

All of these scatterplots show two separate branches of mixing lines. This feature is most obvious in the H2O-O3-correlation. Here, one mixing line connects the tropopause with around 40 ppmv H2O and 100 ppbv O3 and the LMS with decreasing H2O (down to 10 ppmv) at 400 ppbv O3. This mixing line includes also the measured sulfate anomaly and was observed over Northern Germany (see Figs. 7 and C4). The second mixing line is not so pronounced and starts at dryer air masses with 20 ppmv H2O and only goes up to 200 ppbv O3. These observations were made later on the flight over Southern Germany (see also Figs. 7 and C4).

*24. Line 234, "0.01 ppbv", 0.1 ppbv?*

We are sorry for the mistake and corrected it to 0.1 ppbv.

*25. Line 234, "gas-to-particle conversion", It could also be due to the removal process*

Conversion via SO2 to H2SO4, and thereby to particles, is the main removal process of SO2 in the lowermost stratosphere (e.g., Kremser et al., 2016). We agree with the reviewer thatother removal processes for SO2 might also play a role, but we would not expect a steady decrease of SO2 along the mixing line if SO2was removed by other processes. The interesting observation is that we see a continuous decrease of SO2 along the mixing line and at the same time the increase of sulfate aerosol. Considering other removal processes for SO2 there would not be a source for additional particulate sulfate in the ExTL orginating from the troposphere.

*26. Line 272 to 273, "this is ---the stratosphere", This is just a possibility, the link between observations and eruptions is very weak*

We agree that the time span between the eruption and the observation is rather large (7 weeks), but we also would like to point out that the HYSPLIT dispersion calculations show entrainment of the volcano into the stratosphere, especially along the jet stream (see Figs. D1, 10 and B2). Thus, we think that the link between observations and eruptions is not very weak but a reasonable assumption.

*27. Line 275, "seven weeks after the eruption", lack of evidence for this*

We do not have direct evidence, but as explained in our answer to Comment 26, we have reasonable arguments from satellite observations and HYSPLIT trajectories to relate the observations to the volcanic eruption.

*28. Line 289 to 290, "During this --- the stratosphere", In Figures 4, 8, and 9, there seems to be such a transmission line for air masses from the upper troposphere to the stratosphere. But the line's constituent points come from measurements at different locations, elevations and times, meaning they could come from completely different source areas.*

We did not show the map for this specific flight, but in our analysis of RF01, we have two different stratospheric measurement regions and these two regions are also connected with the two different mixing lines, described in Comment 23. The measurement region with high sulfate aerosol is measured over Northern Germany in the vicinity of the jet stream with O3 mixing ratios up to 400 ppbv. The second measurement region is over Southern Germany under subtropical influence. In this region there are very small amounts of sulfate aerosol (approx. 0.1 ppbv), low O3 mixing ratios (< 200 ppbv) and potential temperatures > 350 K. Therefore, we can say that the mixing lines observed especially in Figs. 8 and 9 can be assigned to the different measurement regions probed during this flight. Figure 4 shows the campaign overview in general and not one specific flight.

We added the subfigures to the appendix (Fig. C4) and mentioned them in the main text as described in our answer to Comment 23:

All of these scatterplots show two separate branches of mixing lines. This feature is most obvious in the H2O-O3-correlation. Here, one mixing line starts at around 40 ppmv H2O and

100 ppbv O3 and goes up to 10 ppmv H2O at 400 ppbv O3. This mixing line includes also the measured sulfate anomaly and was observed over Northern Germany (see Figs. 7 and C4). The second mixing line is not so pronounced and starts at dryer air masses with 20 ppmv H2O and only goes up to 200 ppbv O3. These observations were made later on the flight over Southern Germany (see also Figs. 7 and C4).

29. *Line 291, "quasi co-located DLR-Falcon", please elaborate*

In this case-study DLR-Falcon probed the same measurement region as HALO (shown in Fig. C2) with an almost similar flight pattern (flying north-east to south-west and back). The reason for referring to this as "quasi co-located" is that the flight pattern has a time shift of 40 minutes. In the general meteorological situation, this time shift of 40 minutes plays no major role, because both aircraft flew parallel to the axis of the jetstream.

30. *Figure 2, "subset", please define subset*

We generated the subset by extracting the stratospheric data after the method described from line 133 onwards.

We changed the caption of Fig. 2 to:

… The complete stratospheric data set holds 2049 data points which represent 100 % of this subset.

31. *Figure 8, It would be better to add a discussion of why higher potential temperatures occur when ozone concentrations are below 200 ppbv.*

These measurements are conducted in a different meteorological regime during the measurement flight. These potential temperatures occur over Southern Germany under subtropical influence and are not connected to the part of the sulfate anomaly.

See also Comment 28.